# The grin of Cheshire cat resurgence from supersymmetric localization

**Daniele Dorigoni and Philip Glass**

Centre for Particle Theory & Department of Mathematical Sciences, Durham University,
Lower Mountjoy Stockton Road, Durham DH1 3LE, UK

## Abstract

First we compute the $S^2$ partition function of the supersymmetric $\mathbb{CP}^{N-1}$ model via localization and as a check we show that the chiral ring structure can be correctly reproduced. For the $\mathbb{CP}^1$ case we provide a concrete realisation of this ring in terms of Bessel functions. We consider a weak coupling expansion in each topological sector and write it as a finite number of perturbative corrections plus an infinite series of instanton-anti-instanton contributions. To be able to apply resurgent analysis we then consider a non-supersymmetric deformation of the localized model by introducing a small unbalance between the number of bosons and fermions. The perturbative expansion of the deformed model becomes asymptotic and we analyse it within the framework of resurgence theory. Although the perturbative series truncates when we send the deformation parameter to zero we can still reconstruct non-perturbative physics out of the perturbative data in a nice example of Cheshire cat resurgence in quantum field theory. We also show that the same type of resurgence takes place when we consider an analytic continuation in the number of chiral fields from $N$ to $r \in \mathbb{R}$. Although for generic real $r$ supersymmetry is still formally preserved, we find that the perturbative expansion of the supersymmetric partition function becomes asymptotic so that we can use resurgent analysis and only at the end take the limit of integer $r$ to recover the undeformed model.

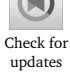

## Contents



# 1 Introduction

Dating back to 1952, an old argument by Dyson [1] suggests that, despite its acclaimed and experimentally confirmed success, the perturbative expansion of QED must have vanishing radius of convergence. The reason for this lack of convergence lies in the asymptotic nature of perturbation theory usually attributed to the rapid factorial growth of Feynman diagrams [2, 3]. Generically, in the absence of magic cancellations, we expect that all the diagrams of a given order will contribute somewhat equally, so that when we sum over all of them we will obtain just from combinatorics a factorial growth for the perturbative coefficients. The asymptotic behaviour of the perturbative series is something very general, deeply rooted within the singular nature of perturbation theory, and an extremely recurrent feature (rather than a bug) present not only in quantum field theory but also in quantum mechanics [4–6] and in string theory [7, 8].

Ecalle resurgence theory [9] is the perfect mathematical framework to address the problem of resummation of asymptotic series. If we only focus on perturbation theory we do not quite get a unique physical answer but rather a family of different analytic continuations. The reason behind this is that perturbation theory is not the end of the story; these ambiguities in resummation generate new non-analytic, i.e. non-perturbative, contributions. Resurgence theory tells us how the global properties of the full solution are intimately linked to these ambiguities [9–13]. Our series expansion has to be replaced by a transseries expansion in which we add on top of the formal power series in the coupling constant these new exponentially suppressed, non-perturbative terms accompanied with their own formal power series. Resurgence theory tells us in practice how to decode from the perturbative data the non-perturbative pieces necessary to construct a unique resummed physical observable: it is possible to disentangle from the perturbative coefficients the fluctuations around different non-perturbative saddle points and vice-versa.

This constructive resurgence program is a very powerful method allowing us to reconstruct non-perturbative physics from perturbative data but ultimately it relies on the asymptotic nature of the perturbative coefficients. However there exist interesting theories for which magic cancellations between diagrams do take place, effectively making perturbation theory a convergent expansion or even better cases for which there are only a finite number of non-vanishing perturbative coefficients. For this class of "special" theories it seems impossible that we can extract non-perturbative information from perturbation theory via a straightforward use of the resurgence program as we do not even have an asymptotic series to begin with.

One might think that, due to cancellations between bosons and fermions, supersymmetric theories would be the perfect candidates for this "good" but "bad" scenarios, however just requiring the theory to be supersymmetric is not a guarantee of a convergent perturbative expansion for every physical observable. In [14, 15] the authors considered different supersymmetric theories in 3 and 4 dimensions (see also [16, 17]) and analysed in great details the weak coupling expansion of particular observables obtained from supersymmetric localization (see [18] for a pedagogical introduction to localization). Despite supersymmetry the authors showed that the perturbative expansions of the considered observables in 4-d $\mathcal{N} = 2$ super Yang-Mills (SYM) were asymptotic but Borel summable, a consequence of the absence of neutral bions configurations as argued in [19, 20]. However using resurgent calculus the authors of [15] were able to extract important non-perturbative information from the perturbative data, although a semi-classical interpretation in terms of microscopic physics for some of these non-perturbative effects is still missing, while for the 3-d $\mathcal{N} = 2$ case discussed in [17] the semi-classical origin of these non-perturbative contributions was very recently understood [21] in terms of complexified supersymmetric solutions.

If we consider $\mathcal{N} = 4\,SU(N)$ SYM in the planar limit the situation changes slightly as we can compute exact quantities using integrability and, thanks to the large number of supersymmetries, the weak coupling expansions of various physical quantities, for example the cusp anomalous dimension [22] and the dressing phase [23], have indeed finite radius of convergence. However not everything is lost from the resurgence point of view since it now happens that the strong coupling expansions of these two observables give rise to asymptotic series, see [22] and [24] respectively. The full resurgence machinery can be then applied to the strong coupling side of planar $\mathcal{N} = 4$ SYM to obtain the complete transseries for the cusp anomaly [25, 26] and the dressing phase [27] leading to important implications for weak/strong coupling interpolation with the stringy $AdS_5 \times S^5$ side, although the semi-classical origin of the non-perturbative effects predicted in [27] is still somewhat mysterious.

The strong coupling side of planar $\mathcal{N} = 4\,SU(N)$ SYM can also be studied within the context of the AdS/CFT correspondence. In particular it was realised in [28] that the hydrodynamic gradient series for the strongly coupled $\mathcal{N} = 4$ super Yang-Mills plasma is only an asymptotic expansion leading to the works [29–31] dealing with resurgence and resummation issues in the fluid context of $AdS_5/CFT_4$.

There are however cases for which we only have access to a convergent weak coupling expansion but we do nonetheless expect non-perturbative physics to be present and for which we do not know an easy way to tackle the strong coupling side with the hope to be able to apply resurgence there. Perhaps the most emblematic example of this sort can be found in supersymmetric quantum mechanics [32] where we can construct simple models for which the ground state energy is zero to all orders in perturbation theory but we do expect non-perturbative physics to play a role. It would seem that in these cases perturbative and non-perturbative data cannot possibly have anything in common with one another, contrary to what usually advertised in the resurgence program. The authors of [33, 34] started precisely from this puzzle and considered two very simple supersymmetric quantum mechanics: the double Sine-Gordon (DSG) and the tilted double well (TDW). The DSG ground state energy has a trivial perturbative expansion and a normalizable ground state, i.e. susy is preserved and $E_0 = 0$ exactly, however the system has real instantons that somehow do not give rise to the expected exponentially suppressed contributions. For the TDW the ground state energy in perturbation theory still vanishes but the model does not have a supersymmetric ground state and its energy should be lifted non-perturbatively although the model does not possess real non-perturbative saddles.

The solutions to both puzzles come from a particular realisation of resurgent theory that the authors of [34] named Cheshire cat resurgence because very much like the magical Won-

derland creature, the lingering grin of resurgence can be still seen from perturbation theory even when its entire body has completely disappeared. The solution is as elegant as simple; we just need to break slightly supersymmetry by declaring that the fermion number in each superselection sector is not an integer anymore but a complex parameter $\zeta$. Once the fermion number is an arbitrary parameter the perturbative expansion of the ground state energy in both systems becomes immediately asymptotic; the body of the cat has appeared once more.

In the deformed DSG case we can apply standard resurgent calculus and obtain a complete transseries expression for the ground state energy that contains not just the perturbative series but also contributions from real as well as complex saddle points [35, 36]. As we send the deformation parameter $\zeta \to 0$ the perturbative series truncates, the contribution coming from the real and complex bions cancel one another because of an hidden topological angle [37], and the ground state energy is exactly zero thanks to the topological quantum interference[1] between different saddle contributions [39]. The role of complex saddles is crucial for this cancellation and their contribution can be really obtained from a semi-classical calculation [40, 41]. However these results have not been yet compared against the predictions coming from resurgent analysis of [42], in which the exact same deformed DSG model is obtained from dimensionally reducing the two dimensional $SU(2)$ $\eta$-deformed principal chiral model and for which the complex bions can be promoted to soliton solutions in the complexified QFT.

A similar story holds for the deformed TDW case: when the deformation parameter $\zeta$ is non zero the perturbative expansion is asymptotic and we can use resurgent analysis to construct from the perturbative data the contribution of the complex bions to the ground state energy. As we send $\zeta \to 0$ the perturbative expansion reduces to zero, while the complex bions remain, as there are no real bions to cancel them, producing non-perturbative contributions to the ground state energy, i.e. supersymmetry is indeed broken. Even if the perturbative expansion truncates both in DSG and TDW we can still use Cheshire cat resurgence to extract non-perturbative physics from perturbative data.

In the present paper we apply the same idea to a two-dimensional quantum field theory. We consider the $\mathbb{CP}^{N-1}$ model, whose resurgent properties have been studied in [43, 44] (see also the recent [45] for connections with 4-d physics), written as a two-dimensional gauged linear sigma model (GLSM) with $\mathcal{N} = (2, 2)$ supersymmetry. The $S^2$ partition function of this model can be computed exactly via localization [46, 47] and its weak coupling expansion can be decomposed as an infinite sum over topological sectors. Each topological sector corresponds to a column in the resurgence triangle [44] and can be written as a perturbative piece plus an infinite tower of non-perturbative terms corresponding to instanton-anti-instanton events, each one of them multiplied by its own perturbative series of fluctuations around it. Due to the supersymmetric nature of the observable under investigation every one of these perturbative series truncates after a finite number of terms, so it would seem that the resurgence program does not allow us to reconstruct the whole column in the resurgence triangle, i.e. the non-perturbative instanton-anti-instanton corrections, from the perturbative expansion in a given topological sector. Following the works of [33, 34] we deform the localized theory by introducing an unbalance, $\Delta = N_f - N_b$, between the number of fermions and bosons present in the theory and immediately we see the full transseries, the body of the Cheshire cat resurgence, popping out. Whenever $\Delta$ is an integer all the perturbative expansions truncate, suggesting that our deformation corresponds to the insertion of some supersymmetric operator; this is very similar to the case $\zeta$ integer and quasi-exact solvability considered in [33, 34]. However as soon as $\Delta$ is kept generic the perturbative expansion becomes asymptotic and we can fully reconstruct the non-perturbative physics out of it. Only at the very end we remove the deformation by considering $\Delta \to 0$ and reconstruct the full supersymmetric result from the

---

[1]In [38] the authors presented a realisation of the same effect in a very nice and simple example involving Bessel functions. We thank Gerald Dunne for discussions on this point.

perturbative data providing a nice example of Cheshire cat resurgence in a supersymmetric quantum field theory.

We further show that a similar structure can be obtained from a more supersymmetric deformation of the model that amounts to an analytic continuation in the number of chiral multiplets[2] from $N \in \mathbb{N}$ (the same $N$ of $\mathbb{CP}^{N-1}$) to a real (or complex) number $r$. Unlike what happens when we introduce $\Delta$, formally in the presence of this deformation the observable under consideration remains supersymmetric. However as soon as $r$ is kept generic, i.e. non-integer, the perturbative expansion becomes asymptotic. We can apply resurgent analysis to reconstruct non-perturbative information from the perturbative data eventually sending $r \to N \in \mathbb{N}$ to recover the original supersymmetric results.

The paper is organised as follows. We first introduce in Section 2 a few generalities about the supersymmetric formulation of the $\mathbb{CP}^{N-1}$ as a gauged linear sigma model (GLSM) and subsequently use localization to compute the partition function of the model when put on $S^2$. As a check in Section 3 we show that the partition function does indeed reproduce the correct twisted chiral ring structure and comment on its connection with the topological-anti-topological partition function. Due to the supersymmetric nature of the observable under consideration we find a perturbative expansion which is far from asymptotic: the perturbative coefficients actually truncate after finitely many orders. For this reason, in Section 4 we deform the theory by introducing an unbalance between the number of bosons and fermions present in the model thus effectively breaking supersymmetry. The deformation considered has a dramatic effect: the perturbative coefficients are not finite in number anymore and perturbation theory becomes an asymptotic expansion. In Section 5 we apply the full machinery of resurgent analysis to the deformed model, where we also show how the intricate set of resurgent relations between the perturbative and non-perturbative sectors survives as we send the deformation parameter to zero. In Section 6 we show that a similar structure can be obtained also when considering a more supersymmetric (at least formally) deformation studying the $\mathbb{CP}^{r-1}$ model defined via analytic continuation from $N \to r \in \mathbb{R}$. We finally conclude in Section 7.

## 2  Supersymmetric $\mathbb{CP}^{N-1}$ as a GLSM

It is useful to briefly review the gauged linear sigma model formulation of the $\mathbb{CP}^{N-1}$ theory with $\mathcal{N} = (2,2)$ supersymmetry; we refer to [48] for all the details. In 2-d the $U(1)$ gauge multiplet is given by a twisted chiral superfield $\Sigma$ containing a complex scalar $\sigma(x)$, as its lowest component, and a $U(1)$ gauge potential $A_\mu(x)$, plus of course fermions. The theory also contains $N$ chiral superfields $\Phi_i$, $i = 1, ..., N$, each charged $+1$ under the gauge group whose lowest components are the complex scalars $\phi_i(x)$. The parameters of the theory are the gauge coupling $e$, which has the dimension of a mass, a dimensionless Fayet-Iliopoulis (FI) term $\xi$, and a vacuum angle $\theta$, that can be combined in the complex coupling $\tau = i\xi + \frac{\theta}{2\pi}$.

The $D$-term conditions for having a supersymmetric vacuum fix $\sigma(x) = 0$ and force the complex scalars $\phi_i(x)$ to satisfy

$$\sum_{i=1}^{N} |\phi_i(x)|^2 = \xi. \tag{1}$$

At energies much smaller than the gauge coupling $e$ the gauge potential is essentially frozen and becomes non-dynamical. We must then identify field configurations

$$\phi_i(x) \sim e^{i\alpha} \phi_i(x), \qquad \forall i = 1, ..., N. \tag{2}$$

---

[2]We thank Stefano Cremonesi for the origin of this idea.

The two conditions (1-2) are precisely the conditions specifying the sigma model with target space $\mathbb{CP}^{N-1}$, so in the infrared the $\mathcal{N} = (2,2)$ gauged linear sigma model becomes the $\mathbb{CP}^{N-1}$ with coupling constant

$$g^2_{\mathbb{CP}^{N-1}} = \frac{1}{\xi}. \tag{3}$$

In all that follows we will express everything in terms of the FI term $\xi$, so that the weak coupling expansion of the $\mathbb{CP}^{N-1}$ model will correspond to the regime $\xi \gg 1$, while the strong coupling expansion will be $\xi \sim 0$.[3]

## 2.1 Supersymmetric partition function on $S^2$

In [46, 47] the authors studied the Euclidean path integral of two-dimensional $\mathcal{N} = (2,2)$ theories with vector and chiral multiplets, placed on a round sphere $S^2$. In the $S^2$ theory the authors constructed a supercharge $\mathcal{Q}$ whose square is a bosonic symmetry and used localization techniques to show that the path integral only receives contribution from classical configurations that are fixed points of $\mathcal{Q}$ and from small quadratic fluctuations around them, i.e. *one-loop determinants*. This set of fixed points is generically discrete or with finite dimension so the path integral reduces dramatically to a sum over topological sectors of ordinary integrals over the Cartan subalgebra of the gauge group dressed by one-loop determinants. We refer to the original works [46, 47] for all the details in the computations and to [49] for a recent review on supersymmetric localization in two dimensions.

We can specialise the work of these authors to the case of a $U(1)$ gauge theory with a FI parameter $\xi$, a $\theta$-term, and with $N_f = N$ chiral multiplet with charge $+1$ and no multiplets with charge $-1$. In absence of twisted masses the localized partition function can be then written as:

$$Z_{\mathbb{CP}^{N-1}} = \sum_{B \in \mathbb{Z}} e^{-i\theta B} \int_{-\infty}^{+\infty} \frac{d\sigma}{2\pi} e^{-4\pi i \xi \sigma} \left( \frac{\Gamma(-i\sigma - B/2)}{\Gamma(1 + i\sigma - B/2)} \right)^N, \tag{4}$$

where the full path integral is reduced to a sum over topological sectors with quantized magnetic flux $B$ times an ordinary integral over the lowest component of the twisted chiral field $\Sigma$ constrained to take the constant value $\sigma(x) = \sigma$ over which we integrate. The first term in the integrand corresponds to the classical action evaluated on shell while the second term in parenthesis comes precisely from the one-loop determinants.

The gamma function at the numerator has poles at locations $\sigma = \sigma_k = -i(k - B/2)$ with $k \in \mathbb{N}$. However for $B \geq 0$ and $k < B$ the integrand is regular because the pole from the numerator is cancelled by the pole from the gamma function in the denominator. For this reason the poles of the integrand are at locations $\sigma = \sigma_k = -i(k + |B|/2)$, and the zeroes are at $\sigma = \sigma_n^{(0)} = +i(n + 1 + |B|/2)$, in particular for $B = 0$ the integrand has a pole at $\sigma = 0$ that has to be included (see [46, 47]) and the integration contour is understood as circling around the pole at the origin in the upper-half complex $\sigma$ plane, see Figure 1.

To evaluate the integral we notice that we can close the contour of integration in the lower-half complex $\sigma$ plane since for $|\sigma| \to \infty$ with $\text{Im}\,\sigma < 0$ the one-loop determinant provides a converging factor, so that we can then rewrite the integral as a sum of residues at the $N^{th}$ order pole locations $\sigma = \sigma_k = -i(k + |B|/2)$. The residue of an $N^{th}$ order pole can be computed as

$$2\pi i \operatorname{Res}_{z=z_0} f(z) = \frac{2\pi i}{\Gamma(N)} \frac{d^{N-1}}{d\alpha^{N-1}} \left[ f(z_0 + \alpha)\alpha^N \right]_{\alpha=0} \tag{5}$$

---

[3]From our analysis we will be also able to consider $\xi \to -\infty$, however this regime does not directly relate to the geometric $\mathbb{CP}^{N-1}$ phase. The reason is that as soon as $\xi < 0$ the $D$-term equation (1) cannot be solved anymore and one needs to use the mirror Landau-Ginzburg theory, see [48, 50]

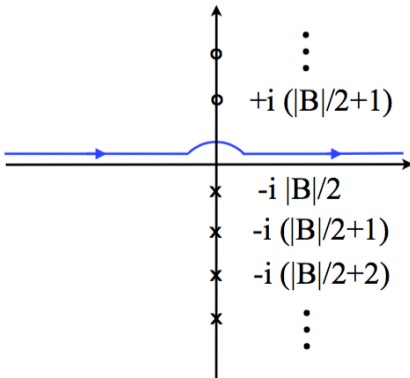

Figure 1: Location of the poles (negative imaginary axis) and of the zeroes (positive imaginary axis) of the one-loop determinant. The contour of integration is deformed in the upper-half plane to avoid the pole at $\sigma = 0$ present in the $B = 0$ case.

and replacing $\alpha \to i\alpha$ we can write the partition function as

$$Z_{\mathbb{CP}^{N-1}} = \frac{1}{(N-1)!} \sum_{B \in \mathbb{Z}} \sum_{k=max(0,B)}^{\infty} \frac{d^{N-1}}{d\alpha^{N-1}} \left[ \alpha^N \frac{\Gamma(-k+\alpha)^N}{\Gamma(1+k-B-\alpha)^N} e^{-4\pi\xi(k-B/2-\alpha)} e^{-i\theta B} \right]_{\alpha=0}.$$ (6)

We can rewrite the above formula introducing the parameter $t = e^{2\pi i \tau} = e^{-2\pi\xi + i\theta}$ written in terms of the complex coupling $\tau = i\xi + \frac{\theta}{2\pi}$ and after simple manipulations with sums indices we get:

$$Z_{\mathbb{CP}^{N-1}} = \frac{1}{(N-1)!} \sum_{n=0}^{\infty} \sum_{m=0}^{\infty} \frac{d^{N-1}}{d\alpha^{N-1}} \left[ \left( \frac{(-1)^n \pi\alpha / \sin(\pi\alpha)}{\Gamma(1+n-\alpha)\Gamma(1+m-\alpha)} \right)^N t^{n-\alpha} \bar{t}^{m-\alpha} \right]_{\alpha=0},$$ (7)

where we also made use of the formula $\Gamma(z)\Gamma(1-z) = \frac{\pi}{\sin(\pi z)}$. We can introduce the regularised generalised hypergeometric function

$${}_1\tilde{F}_N(a; b_1, b_2, ..., b_N | z) = \frac{1}{\Gamma(b_1)\Gamma(b_2)\cdots\Gamma(b_N)} \sum_{n=0}^{\infty} \frac{(a)_n}{(b_1)_n (b_2)_n \cdots (b_N)_n} \frac{z^n}{n!}$$ (8)

where $(a)_n$ denotes the Pochhammer symbol, and our partition function can be written in the compact form:

$$Z_{\mathbb{CP}^{N-1}} = \frac{1}{(N-1)!} \frac{d^{N-1}}{d\alpha^{N-1}} \left[ \left( \frac{\pi\alpha}{\sin(\pi\alpha)} \right)^N (t\bar{t})^{-\alpha} {}_1\tilde{F}_N(1; 1-\alpha, ..., 1-\alpha | (-1)^N t) \right.$$

$$\left. {}_1\tilde{F}_N(1; 1-\alpha, ..., 1-\alpha | \bar{t}) \right]_{\alpha=0}.$$ (9)

It is useful to rewrite the above equation as a sum over instanton sectors each one weighted by the instanton counting parameter $\exp(-2\pi\xi|B| + i\theta B)$ where $B \in \mathbb{Z}$ denotes the instanton number, or equivalently the magnetic flux as above. To this end we can go back to equation (6) and by isolating the instanton counting parameter we obtain

$$Z_{\mathbb{CP}^{N-1}} = \sum_{B \in \mathbb{Z}} e^{-2\pi\xi|B| + i\theta B} \zeta_B(N, \xi),$$ (10)

where the Fourier mode $\zeta_B(N,\xi)$ takes the form

$$\zeta_B(N,\xi) = \frac{(-1)^{NB\,\theta(B)}}{(N-1)!} \sum_{k=0}^{\infty} e^{-4\pi\xi k} \frac{d^{N-1}}{d\alpha^{N-1}} \left[ \left( \frac{(-1)^k \pi\alpha/\sin(\pi\alpha)}{\Gamma(1+k-\alpha)\Gamma(1+k+|B|-\alpha)} \right)^N e^{4\pi\xi\alpha} \right]_{\alpha=0},$$
(11)

with $\theta(B)$ the Heaviside function.

The equations (10)-(11) are very suggestive: the supersymmetric localized partition function for the $\mathbb{CP}^{N-1}$ model on $S^2$ takes the form of an infinite series over instantons sectors, each one of them denoted by an integer $B \in \mathbb{Z}$ and weighted by the instanton counting parameter $\exp(-2\pi\xi|B| + i\theta B)$. Each $B$-instanton sector produces a contribution $\zeta_B(N,\xi)$, function only of the Fayet-Iliopoulos term $\xi$, i.e. the coupling constant, and not of the $\theta$ angle. Every Fourier mode $\zeta_B(N,\xi)$ gives rise to a purely perturbative piece, i.e. the $k=0$ term in (11), plus an infinite sum over exponentially suppressed terms of the form $e^{-4\pi\xi k}$ with $k \in \mathbb{N}^\star$, corresponding to instantons-anti-instantons events, each one of them multiplied by a perturbative expansion. Fixing the instanton number $B$ corresponds to fixing the column in the resurgence triangle diagram of [44] so that $\zeta_B(N,\xi)$ can be interpreted as the transseries containing the perturbative part plus all the instantons-anti-instantons corrections, together with their own perturbative series, on top of a $B$-instanton event.

At this stage we would like to apply resurgent analysis within each instanton sector, i.e. studying separately each transseries $\zeta_B(N,\xi)$ (11) seen as some suitably defined analytic function in some wedge of the complex $\xi$-plane. However it is simple to see that the coefficient of each $e^{-4\pi\xi k}$ term in the infinite sum (11) is actually a polynomial of degree $N-1$ in $\xi$ meaning that both the perturbative expansion around a $B$-instanton event and the perturbative expansions around $k$ instantons-anti-instantons on top of the $B$-instanton event are all entire functions of $\xi$. For a generic observable in a generic field theory we would expect all of these perturbative expansions to be asymptotic series rather than finite degree polynomials. The reason for this truncation is clearly the supersymmetric nature of the quantity under consideration. Being an observable protected by supersymmetry we expect only the first few orders in perturbation theory not to vanish. The same goes for the perturbative expansion on top of non-trivial but still supersymmetric saddles. We are then left with the question: in these lucky situation where the perturbative expansion truncates after a finite number of terms can resurgent analysis tell us anything at all about the non-perturbative completion of the physical observable? At first sight this would seem unlikely, how can an entire function tell you anything about non-perturbative terms? However we will shortly see that Cheshire cat resurgence is at play here: when we focus on this supersymmetric quantity the cat seems to have disappeared but its footprints can still be seen!

## 2.2 The $\mathbb{CP}^1$ case

Instead of working with general $N$ in this Section we specialise equation (9) to the case $N=2$ so that we can give shorter and less cluttered equations, the discussions however can be repeated for the general case. To compute the partition function on $S^2$ for the $\mathbb{CP}^1$ model we simply take (9) and substitute $N=2$:

$$Z_{\mathbb{CP}^1} = \frac{d}{d\alpha} \left[ \left( \frac{\pi\alpha}{\sin(\pi\alpha)} \right)^2 (t\bar{t})^{-\alpha} \,_1\tilde{F}_2(1;1-\alpha,1-\alpha|t) \,_1\tilde{F}_2(1;1-\alpha,1-\alpha|\bar{t}) \right]_{\alpha=0}.$$
(12)

When the derivative with respect to $\alpha$ does not act on the hypergeometric function we obtain terms of the form $\,_1\tilde{F}_2(1;1,1|t) = I_0(2\sqrt{t})$, where $I_0$ denotes the modified Bessel function of 0-th order, whilst when the derivative acts on the hypergeometric parameters, we obtain terms

of the form

$$\frac{d}{db_1}\,_1F_2(a;b_1,b_2|z) = \psi(b_1)\,_1F_2(a;b_1,b_2|z) - \sum_{k=0}^{\infty}\frac{(a)_k\,\psi(k+b_1)}{(b_1)_k(b_2)_k}\frac{z^k}{k!}\,, \tag{13}$$

where $\psi(x)$ denotes the digamma function and $-\psi(1) = \gamma$ gives the Euler-Mascheroni constant. So we obtain

$$Z_{\mathbb{CP}^1} = -\log(t\bar{t})\,_0F_1(1|t)\,_0F_1(1|\bar{t}) - 2\,_0F_1(1|\bar{t})\frac{d}{db}\,_0\tilde{F}_1(b|t)|_{b=1} - 2\,_0F_1(1|t)\frac{d}{db}\,_0\tilde{F}_1(b|\bar{t})|_{b=1}\,,$$

and changing variable $b = 1 + a$ we can write

$$_0\tilde{F}_1(1+a|z) = \frac{1}{(\sqrt{z})^a}I_a(2\sqrt{z})\,,$$

that together with the relation

$$\frac{d}{da}I_a(z)|_{a=0} = -K_0(z)$$

brings us to the final form

$$Z_{\mathbb{CP}^1} = 2\left(I_0(2\sqrt{t})K_0(2\sqrt{\bar{t}}) + K_0(2\sqrt{t})I_0(2\sqrt{\bar{t}})\right)\,. \tag{14}$$

It is also useful to specialise the Fourier mode decomposition (10) to the $\mathbb{CP}^1$ case

$$Z_{\mathbb{CP}^1} = \sum_{B\in\mathbb{Z}}e^{-2\pi\xi|B|+i\theta B}\zeta_B(2,\xi)\,, \tag{15}$$

with

$$\zeta_B(2,\xi) = \sum_{k=0}^{\infty}e^{-4\pi\xi k}(4\pi\xi)^2\left[\frac{1}{[k!\,(k+|B|)!]^2}(4\pi\xi)^{-1} + \frac{2\psi(k+1)+2\psi(k+|B|+1)}{[k!\,(k+|B|)!]^2}(4\pi\xi)^{-2}\right]$$

$$= \frac{4\pi\xi - 2\gamma + 2\psi(|B|+1)}{|B|!^2} + \frac{4\pi\xi + 2(1-\gamma)) + 2\psi(|B|+1)}{[(|B|+1)!]^2}\times e^{-4\pi\xi} + \mathrm{O}(e^{-8\pi\xi})\,. \tag{16}$$

Since in what follows we will mostly consider the $B = 0$ sector we can specialise the above equation even further obtaining the very simple expression

$$\zeta_0(2,\xi) = \sum_{k=0}^{\infty}e^{-4\pi\xi k}(4\pi\xi)^2\left[\frac{1}{(k!)^4}(4\pi\xi)^{-1} + \frac{4H_k-4\gamma}{(k!)^4}(4\pi\xi)^{-2}\right]\,, \tag{17}$$

where $H_k = \psi(1+k) + \gamma$ denotes the $k^{th}$ harmonic number. Equation (17) can be written in terms of Meijer $G$ function and it is neither asymptotic in the weak coupling regime $\xi \to \infty$ nor in the strong coupling one $\xi \to 0$.

As mentioned above each topological sector can be written as a purely perturbative expansion, given by a very simple degree 1 polynomial in $\xi$, plus an infinite tower of instanton-anti-instanton events, weighted by $e^{-4\pi\xi}$, each one of them accompanied by a simple perturbative expansion given by a different degree 1 polynomial in $\xi$. Due to the supersymmetric nature of the observable under consideration perturbation theory is not asymptotic at all, it actually truncates after finitely many terms so that there is no need to apply Borel resummation and the perturbative expansion appears to be completely oblivious of the non-perturbative sectors. We will see later on that this is precisely an example of Cheshire cat resurgence at play in quantum field theory.

# 3   Chiral ring structure

Having obtained the partition function for $\mathbb{CP}^1$ (14) and more generically for $\mathbb{CP}^{N-1}$ (9) we can compute the chiral ring for these models, see [50]. For $\mathbb{CP}^{N-1}$ the ring is generated by one element $\Sigma$ that at the classical level satisfies $\Sigma^N = 0$, but receives instantons corrections and it gets modified to

$$\Sigma^N = \Lambda^N_{\mathbb{CP}^{N-1}}\,, \tag{18}$$

where $\Lambda_{\mathbb{CP}^{N-1}} = \mu e^{-2\pi\xi/N + i\theta/N} = \mu\, t^{1/N}$. The top component of $\Sigma$ is related to the action itself via:

$$S = \log t \int d^2x\, d^2\theta\, \Sigma + h.c. = 2\pi i\,\tau \int d^2x\, d^2\theta\, \Sigma + h.c. \tag{19}$$

so we can generate the full chiral ring by considering[4]

$$\langle \Sigma^n \bar{\Sigma}^m \rangle = \frac{1}{Z_{\mathbb{CP}^{N-1}}} (t\partial_t)^n (\bar{t}\partial_{\bar{t}})^m Z_{\mathbb{CP}^{N-1}} = \frac{1}{(2\pi i)^n (-2\pi i)^m} \partial_\tau^n \partial_{\bar{\tau}}^m \log Z_{\mathbb{CP}^{N-1}}\,. \tag{20}$$

## 3.1   Chiral ring for $\mathbb{CP}^1$

Let us start with $N = 2$ and use (14) to compute $\langle\Sigma\rangle$ obtaining:

$$\langle\Sigma\rangle = \frac{t}{Z_{\mathbb{CP}^1}} \partial_t Z_{\mathbb{CP}^1} = 2\sqrt{t}\left(I_1(2\sqrt{t})K_0(2\sqrt{t}) - K_1(2\sqrt{t})I_0(2\sqrt{t})\right)/Z_{\mathbb{CP}^1}\,. \tag{21}$$

We can easily compute $\langle\Sigma^2\rangle = 1/Z_{\mathbb{CP}^1}\, t\,\partial_t(t\,\partial_t Z_{\mathbb{CP}^1})$ and making use of the relations for the modified Bessel:

$$I'_\nu(z) = I_{\nu-1}(z) - \frac{\nu}{z}I_\nu(z)\,,$$
$$K'_\nu(z) = -K_{\nu-1}(z) - \frac{\nu}{z}K_\nu(z)\,, \tag{22}$$

we obtain

$$\langle\Sigma^2\rangle = t = \Lambda^2_{\mathbb{CP}^1}\,, \tag{23}$$

as expected from the chiral ring structure (18). The $S^2$ localized partition function and its derivatives with respect to the the (anti-)holomorphic coupling give rise to a representation of the chiral ring in terms of modified Bessel functions.

It was shown in [51] that in the superconformal case, where the sigma model target space is a Calabi-Yau manifold rather than $\mathbb{CP}^{N-1}$, the supersymmetric localized partition function on the round two-sphere matched precisely the exact Kähler potential on the quantum Kähler moduli space of the Calabi-Yau emerging in the infrared. This means that in the superconformal case the localized partition function coincides with the seemingly unrelated topological-anti-topological construction of Cecotti and Vafa [52].

The model we are considering is however an asymptotically free theory rather than a superconformal one and it is not clear how to relate the two-sphere localized calculations to the $\mathbb{CP}^1$ topological-anti-topological results obtained in [50]. To this end, we first complete the chiral ring (23) (similarly for $\bar{\Sigma}$) and study correlators with multiple $\Sigma$, $\bar{\Sigma}$ insertions obtaining

---

[4]By slight abuse of notation from this point onward we denote insertions of the top component of $\Sigma$ with $\Sigma$ itself.

the functions:

$$\langle\Sigma\rangle = \frac{t}{Z_{\mathbb{CP}^1}}\,\partial_t Z_{\mathbb{CP}^1} = 2\sqrt{t}\left(I_1(2\sqrt{t})K_0(2\sqrt{t}) - K_1(2\sqrt{t})I_0(2\sqrt{t})\right)/Z_{\mathbb{CP}^1}\,, \tag{24}$$

$$\langle\bar\Sigma\rangle = \frac{\bar t}{Z_{\mathbb{CP}^1}}\,\partial_{\bar t} Z_{\mathbb{CP}^1} = 2\sqrt{\bar t}\left(K_0(2\sqrt{\bar t})I_1(2\sqrt{\bar t}) - I_0(2\sqrt{\bar t})K_1(2\sqrt{\bar t})\right)/Z_{\mathbb{CP}^1}\,, \tag{25}$$

$$\langle\bar\Sigma\Sigma\rangle = \frac{t\bar t}{Z_{\mathbb{CP}^1}}\,\partial_t\partial_{\bar t} Z_{\mathbb{CP}^1} = -2\sqrt{t\,\bar t}\left(I_1(2\sqrt{t})K_1(2\sqrt{\bar t}) + K_1(2\sqrt{t})I_1(2\sqrt{\bar t})\right)/Z_{\mathbb{CP}^1}\,. \tag{26}$$

With these functions we can construct the hermitian metric

$$g = \begin{pmatrix} \langle 1\rangle & \langle\bar\Sigma\rangle \\ \langle\Sigma\rangle & \langle\bar\Sigma\Sigma\rangle \end{pmatrix}\,, \tag{27}$$

and note that it is manifestly not diagonal unlike the metric considered in [50]. The reason is that on $S^2$, compared to $\mathbb{R}^2$, we have operators mixing with lower dimensional ones,[5] see [53], in particular $\Sigma$ and $\bar\Sigma$ mix with the identity. The determinant of the matrix $g$ removes this mixing and produces the only relevant correlator for $\mathbb{CP}^1$ given by the connected correlator $\langle\Sigma\bar\Sigma\rangle_C = \langle\Sigma\bar\Sigma\rangle - \langle\Sigma\rangle\langle\bar\Sigma\rangle$. This determinant can be easily computed using the relation

$$I_{\nu+1}(z)K_\nu(z) + I_\nu(z)K_{\nu+1}(z) = \frac{1}{z} \tag{28}$$

arriving at $\det g = -1/Z_{\mathbb{CP}^1}^2$, so the only function we need to consider for the $t\bar t$-equations of [50] is precisely $Z_{\mathbb{CP}^1}$.

It is now a matter of calculation to show that our result does not quite solve the topological-anti-topological equation of [50] but rather satisfies a simple modification of it

$$t\bar t\,\partial_t\partial_{\bar t}\log Z_{\mathbb{CP}^1} = 0 \times t\bar t\, Z_{\mathbb{CP}^1}^2 - \frac{1}{Z_{\mathbb{CP}^1}^2}\,, \tag{29}$$

or using the same notation as [50] we can define $q_0 = -q_1 = \log Z_{\mathbb{CP}^1} + \frac{1}{4}\log|t|^2$ and using the variable $z = 2\sqrt{t}$ (our $t$ corresponds to their $\beta$) we can rewrite (29) as

$$\partial_z\partial_{\bar z}q_0 = 0 \times e^{2q_0} - e^{-2q_0}\,. \tag{30}$$

Had the coefficient of the first term on the right-hand side in (29-30) been 1 instead of 0 we would have found precisely the Toda equation of [50], however we do not know why we obtain this modification. It is possible that because of $UV$ divergences one has to regulate insertions of the composite operator $\Sigma\bar\Sigma$ to correctly reproduce the topological-anti-topological results from supersymmetric localization, or it could also happen that the localization calculation in the non-superconformal case is computing something genuinely different from [50]. These are very interesting questions deserving more studies however they fall outside of (and will not affect) the main message of this paper.

## 3.2 Chiral ring for $\mathbb{CP}^{N-1}$

Starting from equation (9) we want to show that the chiral ring structure $\langle\Sigma^N\rangle = \Lambda_{\mathbb{CP}^{N-1}}^N$ can be obtained from the supersymmetric localized partition function. To this end we need to show that the equation

$$(t\,\partial_t)^N Z_{\mathbb{CP}^{N-1}} = \Sigma^N \tag{31}$$

---

[5]We thank Vasilis Niarchos for discussions on this point.

holds. Instead of working with (9) we can use the power series expansion (7) and when we act with the operator $(t\,\partial_t)^N$ on $Z_{\mathbb{CP}^{N-1}}$ we can commute the derivatives with the series and the only term we have to consider is $t^{n-\alpha}$ for which we obtain the simple action

$$(t\,\partial_t)^N \frac{t^{n-\alpha}}{\Gamma(1+n-\alpha)^N} = \frac{(n-\alpha)^N}{\Gamma(1+n-\alpha)^N} t^{n-\alpha} = \frac{t^{n-\alpha}}{\Gamma(n-\alpha)^N}\,. \tag{32}$$

We can thus shift $n \to n+1$ and obtain

$$(t\,\partial_t)^N Z_{\mathbb{CP}^{N-1}} = t Z_{\mathbb{CP}^{N-1}} +$$
$$+ \frac{1}{(N-1)!} \sum_{m=0}^{\infty} \frac{d^{N-1}}{d\alpha^{N-1}} \left[ \left( \frac{\pi\alpha/\sin(\pi\alpha)}{\Gamma(-\alpha)\Gamma(1+m-\alpha)} \right)^N t^{-\alpha}\, \bar{t}^{m-\alpha} \right]_{\alpha=0}\,, \tag{33}$$

where the second term comes from the $n = 0$ contribution in (9) after we use (32). This second term vanishes because the $1/\Gamma(-\alpha)^N$ term has an $N^{th}$ order zero when $\alpha \to 0$ and at most $N-1$ derivatives with respect to $\alpha$ can act upon it. Hence we obtain the expected chiral ring structure

$$\langle \Sigma^N \rangle = \frac{1}{Z_{\mathbb{CP}^{N-1}}} (t\,\partial_t)^N Z_{\mathbb{CP}^{N-1}} = t = \Lambda^N_{\mathbb{CP}^{N-1}}\,. \tag{34}$$

It would be interesting to construct general correlation functions of the form $\langle \Sigma^n \bar{\Sigma}^m \rangle$ to see if we can find a solution to some modification of the affine Toda equations presented in [50], similar to what we obtained for $\mathbb{CP}^1$ in equation (30), however this is beyond the purpose of the present paper.

The reader should now be convinced that the $S^2$ partition function does indeed capture various physical properties of the supersymmetric $\mathbb{CP}^{N-1}$ model. However due to supersymmetry, the weak coupling expansion does not give rise to any asymptotic series but it does nonetheless contain infinitely many non-perturbative corrections, seemingly defying the resurgence program whose task is to reconstruct non-perturbative information out of perturbative data. In the next Section we will see how to get around these superficial negative results by breaking supersymmetry in a controlled way.

## 4 Away from the supersymmetric point

Since each instanton sector in (11) gives rise, due to supersymmetry, to a convergent rather than an asymptotic expansion it would appear that resurgent analysis cannot be applied in the model at hand. However motivated by the works [33, 34] we decided to modify slightly the localized path integral by unbalancing the number of bosons and fermions in the one-loop determinants so that supersymmetry is broken but in a very tamed manner.

To obtain via supersymmetric localization the partition function presented in (4), after having found the localized critical points one has to compute the one-loop determinant for the quadratic fluctuations around these BPS configurations. For the $\mathbb{CP}^{N-1}$ model the one-loop determinant for the vector multiplet is just 1 while it becomes non-trivial for the chiral multiplet. For a single chiral multiplet the matter one-loop determinant is given by

$$Z_{matter} = \frac{\det \mathcal{O}_\psi}{\det \mathcal{O}_\phi}\,, \tag{35}$$

where $\phi$ and $\psi$ denote respectively the complex scalar and the Dirac fermion in the multiplet

and the one-loop determinants are given by (see [46, 47])

$$\det \mathcal{O}_\phi = \prod_{j=\frac{|B|}{2}}^{\infty} (j - i\sigma)^{2j+1}(j + 1 + i\sigma)^{2j+1}, \tag{36}$$

$$\det \mathcal{O}_\psi = (-1)^{B\theta(B)} \prod_{k=\frac{|B|}{2}}^{\infty} (k - i\sigma)^{2k}(k + 1 + i\sigma)^{2k+2}. \tag{37}$$

As discussed in Section 2, the GLSM realisation of the $\mathbb{CP}^{N-1}$ model contains $N$ chiral multiplets so that the matter one-loop determinant contribution to the partition function amounts to $Z_{matter}^N$. However at this point, in a similar way to the works [33, 34], we want to introduce a small unbalance between the bosonic and fermionic contributions to the matter one-loop determinant by declaring that after having localized on the susy critical points we have $N_f = N$ fermions but only $N_b = N - \Delta$ bosons so that

$$\tilde{Z}_{matter}(\sigma) = \frac{\left(\det \mathcal{O}_\psi\right)^{N_f}}{\left(\det \mathcal{O}_\phi\right)^{N_b}} = Z_{matter}^N \left(\det \mathcal{O}_\phi\right)^{\Delta}, \tag{38}$$

and when $\Delta = N_f - N_b$ vanishes we go back to the undeformed, supersymmetric case.

By using (36-37) we can rewrite $\tilde{Z}_{matter}$ as

$$\tilde{Z}_{matter}(\sigma) = (-1)^{NB\theta(B)} \prod_{j=0}^{\infty} \left(\frac{j+b}{j+a}\right)^{N-\Delta(2i\sigma+1)} \cdot (j+a)^{2\Delta(j+a)} \cdot (j+b)^{2\Delta(j+b)}, \tag{39}$$

where we defined $a = |B|/2 - i\sigma$ and $b = 1 + i\sigma + |B|/2$.

We can use zeta-function regularisation to define these infinite products, see details in Appendix A, and using equations (106)-(111) we obtain a regularised version of the modified matter one-loop determinant

$$\begin{aligned}
\tilde{Z}_{matter}(\sigma) = &\left[(-1)^{B\theta(B)} \frac{\Gamma(-i\sigma + |B|/2)}{\Gamma(1 + i\sigma + |B|/2)}\right]^N e^{-2\Delta\left(2\zeta'(-1)+\zeta'(0)(|B|+1)+|B|^2/4+i\sigma-\sigma^2\right)} \\
&\times \exp\left[\Delta(2i\sigma + 1)\left(\log\Gamma(1 + i\sigma + |B|/2) - \log\Gamma(-i\sigma + |B|/2)\right)\right] \\
&\times \exp\left[-2\Delta\left(\psi^{(-2)}(1 + i\sigma + |B|/2) + \psi^{(-2)}(-i\sigma + |B|/2)\right)\right]. \tag{40}
\end{aligned}$$

One can recognise that the above one-loop determinants are very similar to the effective actions for bosonic and fermionic fields on the hyperbolic manifold $H^2$ used as building blocks to study the strong-coupling expansions for the Wilson loop minimal surfaces in AdS$_5 \times$ S$^5$ (see e.g. [54]). The very same effective actions have been studied in [55] using generalised dyadic identities for the polygamma function to obtain inverse factorial series expansion. It would be interesting to understand how to apply the results of [55] to the current problem.

Using the properties of the gamma function, see [46], one can rewrite the first parenthesis in the above expression to put it back into the form $\Gamma(-i\sigma - B/2)/\Gamma(1 + i\sigma - B/2)$ which appears in the undeformed one-loop determinant as in (4). All the remaining terms have the form $e^{\Delta(\cdots)}$ clearly tending to 1 as $\Delta \to 0$.

It is crucial for what follows to analyse the analytic properties of $\tilde{Z}_{matter}$ as a function of $\sigma$. In the undeformed case, see Figure 1 and the discussion below equation (4), we had poles for $\sigma = -i(k + |B|/2)$ and zeroes for $\sigma = +i(1 + n + |B|/2)$ with $k, n \in \mathbb{N}$. However due to the presence of $\log\Gamma$ and $\psi^{(-2)}$ instead of poles and zeroes we have two branch cuts running along the positive and negative imaginary axis. The functions $\log\Gamma(z)$ and $\psi^{(-2)}(z)$



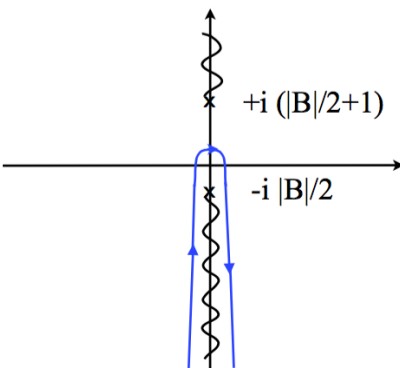

Figure 2: The contour of integration $\mathscr{C}$ comes from $-i\infty - \epsilon$, circles around the branch cut and then goes back to $-i\infty + \epsilon$.

are analytic throughout the complex $z$-plane, except for a single branch cut discontinuity along the negative real axis.[6] The discontinuities of $\log\Gamma$ and $\psi^{(-2)}$ can be easily computed

$$\log\Gamma(-x + i\epsilon) - \log\Gamma(-x - i\epsilon) = -2\pi i (\lfloor x \rfloor + 1), \tag{41}$$

$$\psi^{(-2)}(-x + i\epsilon) - \psi^{(-2)}(-x - i\epsilon) = \pi i (\lfloor x \rfloor + 1)(2x - \lfloor x \rfloor), \tag{42}$$

for $x \geq 0$, where $\lfloor x \rfloor$ denotes the floor of $x$.

We are now in position to study our deformed localized path integral taking the form

$$Z(N, \Delta) = \sum_{B \in \mathbb{Z}} e^{-i\theta B} \int_{\mathscr{C}} \frac{d\sigma}{2\pi} e^{-4\pi i \xi \sigma} \tilde{Z}_{matter}(\sigma), \tag{43}$$

where $\mathscr{C}$ is a suitably defined contour in the complex $\sigma$ plane. In the superymmetric case $\mathscr{C}$ is given by the real line that we subsequently close in the lower-half complex plane collecting the residues from all the poles of the integrand. If we repeat for the case at hand and push the contour of integration in the lower-half complex plane, due to the presence of the branch cut, we end up with a contour $\mathscr{C}$ along the negative imaginary axis, coming from $\sigma = -i\infty - \epsilon$, circling around the origin of the branch cut at $\sigma = -i|B|/2$ and the continuing back to infinity in the direction $\sigma = -i\infty + \epsilon$ as depicted in Figure 2, we will comment later on the analytic properties of these branch cuts.

Note that for any $\Delta \neq 0$ this is just a formal definition[7] since the integrand in (43) behaves as $\tilde{Z}_{matter}(\sigma) \sim \exp[-2\Delta\cos(2\theta)R^2 \log R]$ when $|\sigma| = R \to \infty$, with $\theta = \arg\sigma$ and closing the contour in the lower-half plane will produce a different analytic continuation. However if we insist on taking the contour $\mathscr{C}$ to be the one presented in Figure 2 and consider the integral (43) only as a formal object, we will see that as we send $\Delta \to 0$ everything will be well-defined.[8]

---

[6]In here we use a specific determination of $\log\Gamma$, what Mathematica calls `LogGamma[z]`. The function $\log(\Gamma(z))$ has a more complex branch cut structure.

[7]If one does not want to work with formal objects we can add a quartic twisted superpotential allowing us to close the contour on the imaginary axis. Now everything becomes well defined and convergent so we can check numerically that all the formal equations derived using resurgent analysis are indeed correct, and only at the very end we send this auxiliary quartic coupling to zero.

[8]This situation is similar to the case [56] of bad $\mathcal{N} = 4$ theories in 3-d where it can be shown that the localized matrix integral over the "original" contour of integration diverges but can be regularised by modifying the contour in the complex (fields) space. It is only with this deformed contour of integration that one obtains a well defined integral that can be understood in terms of infrared physics [57]. We thank Stefano Cremonesi for discussions on this point.

Once the contour is fixed we can make the change of variable $\sigma = -iy$ and rewrite (43) as

$$Z(N,\Delta) = \sum_{B \in \mathbb{Z}} e^{-i\theta B} \int_0^\infty \frac{dy}{2\pi i} e^{-4\pi \xi y} \left( \tilde{Z}_{matter}(-iy+\epsilon) - \tilde{Z}_{matter}(-iy-\epsilon) \right). \qquad (44)$$

The integral in the above expression is nothing but the Laplace transform of the discontinuity of $\tilde{Z}_{matter}$ along the negative imaginary axis. This discontinuity starts at $y = |B|/2$, so after shifting $y = x + |B|/2$ we obtain a Fourier mode expansion of the same form as the original one (10)

$$Z(N,\Delta) = \sum_{B \in \mathbb{Z}} e^{-2\pi \xi |B| - i\theta B} \tilde{\zeta}_B(N,\xi,\Delta), \qquad (45)$$

and in each topological sector we can make use of the discontinuities equations (41-42) to obtain

$$\tilde{\zeta}_B(N,\xi,\Delta) = \sum_{k=0}^\infty \int_k^{k+1} \frac{dx}{2\pi i} e^{-4\pi \xi x} \tilde{Z}_{matter}(-ix - i|B|/2 - \epsilon) \left[ e^{-2\pi i \Delta(k+1)(k+|B|+1)} - 1 \right]$$

$$= \sum_{k=0}^\infty \int_k^{k+1} \frac{dx}{2\pi i} e^{-4\pi \xi x} \tilde{Z}_{matter}(-ix - i|B|/2 + \epsilon) \left[ 1 - e^{+2\pi i \Delta(k+1)(k+|B|+1)} \right]. \quad (46)$$

We can rewrite each integral as $\int_k^{k+1} = \int_k^\infty - \int_{k+1}^\infty$ and then shift integration variables so that every integral becomes between $[0,\infty)$, arriving at

$$\tilde{\zeta}_B(N,\xi,\Delta) = \sum_{k=0}^\infty e^{-4\pi \xi k} \int_0^\infty \frac{dx}{2\pi i} e^{-4\pi \xi x} \left[ (-1)^{B\theta(B)} \frac{\Gamma(-x-k)}{\Gamma(1+x+k+|B|)} \right]^N f(x,\Delta)$$

$$\times \exp\left[ -\Delta(2x + 2k + |B| + 1) \log \Gamma(-x - k + i\epsilon) - 2\Delta \psi^{(-2)}(-x - k + i\epsilon) \right]$$

$$\times \exp\left[ \Delta(2x + 2k + |B| + 1) \log \Gamma(x + k + |B| + 1) - 2\Delta \psi^{(-2)}(x + k + |B| + 1) \right]$$

$$\times \left[ e^{-2\pi i \Delta(k+1)(k+|B|+1)} - e^{-2\pi i \Delta k(k+|B|)} \right], \qquad (47)$$

where $f(x,\Delta)$ is an entire function of $x$ that goes to $1$ as $\Delta \to 0$ given by $f(x,\Delta) = \exp[-2\Delta(x^2 + x + c)]$ with $c$ an $x$ independent constant. Note that a similar equation can be straightforwardly derived for $\epsilon \to -\epsilon$.

This equation will be the starting point of our resurgent analysis of the deformed theory: the $B$ instanton sector contribution $\tilde{\zeta}_B(N,\xi,\Delta)$ has been written as the sum over instanton-anti-instanton events, weighted by $e^{-4\pi \xi k}$, each one of them multiplied by the Laplace transform of a function with branch cuts in the directions $\arg x = 0$, coming from the first exponential in the integrand, and $\arg x = \pi$, coming from the second exponential in the integrand, these being the only two *Stokes* directions. As we will shortly see, a weak-coupling expansion of the Laplace integral in (47) will give rise to asymptotic series in $\xi^{-1}$ with $\Delta$ dependent coefficients. Furthermore since $f(x,\Delta)$ is an entire function of $x$ it will not change the asymptotic nature of the perturbative expansion, so for this function we can safely set $\Delta = 0$ and replace $f(x,\Delta) \to f(x,0) = 1$ without modifying the resurgence structure.[9]

We can rewrite (47)

$$\tilde{\zeta}_B(N,\xi,\Delta) = (-1)^{NB\,\theta(B)} \sum_{k=0}^\infty e^{-4\pi \xi k} e^{\pm i\pi k(k+|B|)\Delta} \mathcal{S}_\pm \left[ \Phi_B^{(k)} \right](\xi,\Delta) \qquad (48)$$

---

[9]If we expand for $z$ large the Laplace transform of the product of two functions $\int_0^\infty e^{-xz} f(x)g(x) = \sum_{n=0}^\infty n! c_n z^{-n-1}$ we have that the coefficients $c_n$ are given by the convolution sum $c_n = \sum_{k=0}^n a_{n-k} b_k$ where $f(x) = \sum_{n=0}^\infty a_n x^n$ and $g(x) = \sum_{n=0}^\infty b_n x^n$. This convolution amounts to a change in the definition of coupling constant, i.e. $z^{-1} \to w^{-1} = F(z^{-1})$, and this change of variable is entire when the function $f(x)$ is, so that the resurgence properties remain the same.

where $\mathscr{S}_{\pm}$ denote the lateral Laplace transforms

$$\mathscr{S}_{\pm}\left[\Phi_B^{(k)}\right](\xi,\Delta) = \int_0^{\infty\pm i\epsilon} dx\, e^{-4\pi\xi x}\, x^{-N+\Delta(2k+1+|B|)}\, \Phi_B^{(k)}(x,\Delta), \tag{49}$$

obtained as the limiting case approaching a Stokes line of the directional Borel resummation

$$\mathscr{S}_\theta\left[\Phi_B^{(k)}\right](\xi,\Delta) = \int_0^{\infty e^{i\theta}} dx\, e^{-4\pi\xi x}\, x^{-N+\Delta(2k+1+|B|)}\, \Phi_B^{(k)}(x,\Delta). \tag{50}$$

The Borel transform $\Phi_B^{(k)}(x,\Delta)$ appearing in the above equation can be rewritten from (47) as

$$\Phi_B^{(k)}(x,\Delta) = -\frac{\sin[\pi\Delta(2k+|B|+1)]}{\pi}\left[\frac{(-1)^{k+1}\pi x/\sin(\pi x)}{\Gamma(1+x+k)\Gamma(1+x+k+|B|)}\right]^N$$
$$\times \exp\Big[-\Delta(2x+2k+|B|+1)(\log\Gamma(1-x)-\log((x+1)_k))-2\Delta\big(\psi^{(-2)}(1-x)+$$
$$-\psi^{(-2)}(k+1)-(k+1)(x+k)+k\log k+\sum_{j=1}^k[(x+j)\log(x+j)+(k-j)\log(k-j)]\big)\Big]$$
$$\times \exp\Big[\Delta(2x+2k+|B|+1)\log\Gamma(x+k+|B|+1)-2\Delta\psi^{(-2)}(x+k+|B|+1)\Big], \tag{51}$$

after repeated use of the formulas

$$\log\Gamma(-x\pm i\epsilon) = \log\Gamma(1-x\pm i\epsilon)-\log x\mp i\pi, \tag{52}$$
$$\psi^{(-2)}(-x\pm i\epsilon) = \psi^{(-2)}(1-x\pm i\epsilon)-\psi^{(-2)}(1)+x[\log x-1]\pm i\pi x, \tag{53}$$

valid for $x\geq 0$. For example the purely perturbative contribution $k=0$ in the trivial topological sector $B=0$ can be obtain from the directional Laplace transform of

$$\Phi_0^{(0)}(x,\Delta) = -\frac{(-1)^N\sin(\pi\Delta)}{\pi}\left[\frac{\pi x/\sin(\pi x)}{\Gamma(1+x)^2}\right]^N\exp\Big[2\Delta(x+\psi^{(-2)}(1))\Big]\times \tag{54}$$
$$\exp\Big[\Delta(2x+1)(\log\Gamma(1+x)-\log\Gamma(1-x))-2\Delta\big(\psi^{(-2)}(1+x)+\psi^{(-2)}(1-x)\big)\Big].$$

It is now clear from (51) or (54) that we cannot naively take the limit $\Delta\to 0$ since the overall factor $\sin[\pi\Delta(2k+|B|+1)]$ coming from the discontinuity (46) vanishes. However, as we will shortly see, precisely in this limit this factor multiplies an asymptotic series in inverse powers of $\xi$ with singular coefficients.

From equation (50) we can understand the branch structure introduced by our deformed one-loop determinant (40) that was schematically depicted in Figure 2. In the directional Borel resummation (50) we have split the branches into two separate structures. First we notice the $x^{\Delta(2k+1+|B|)}$ term that for generic $\Delta$ introduces a cut starting from the origin $x=0$. This non-analytic term will serve as a regulator and it will be crucial to properly recover the supersymmetric result from the deformed theory. Secondly the modified Borel transforms, i.e the functions $\Phi_B^{(k)}(x,\Delta)$, have branch cuts starting at $x=+1$ running to $x\to+\infty$ and at $x=-1-|B|$ running to $x\to-\infty$, so that their only singular directions, i.e. their *Stokes lines*, are $\arg x=0$ and $\arg x=\pi$. We will shortly see the consequences of these facts.

## 5 Cheshire cat Resurgence

The first thing we can check from our expansion (48) is that we reproduce the undeformed case (11) or (16) in the case of $\mathbb{CP}^1$, i.e. $N=2$. The key point is that our Borel transform (51)

for $x \sim 0$ behaves as

$$\Phi_B^{(k)}(x, \Delta) \sim -\frac{\sin[(2k+1+|B|)\pi\Delta]}{\pi} \left( \sum_{n=0}^{\infty} c_{B,n}^{(k)}(\Delta) x^n \right), \tag{55}$$

where the coefficients $c_{B,n}^{(k)}(\Delta)$ can be easily obtained from (51). For $\Delta = 0$ these coefficients are simply the Taylor coefficients of the function $\left[ \frac{(-1)^{k+1}\pi x/\sin(\pi x)}{\Gamma(1+x+k)\Gamma(1+x+k+|B|)} \right]^N$:

$$c_{B,0}^{(k)}(0) = \left( \frac{(-1)^{k+1}}{k!(k+|B|)!} \right)^N,$$
$$c_{B,1}^{(k)}(0) = -N[\psi(k+1) + \psi(k+|B|+1)] c_{B,0}^{(k)}(0). \tag{56}$$

So if we consider a weak coupling expansion, i.e. $\xi \to \infty$, of the lateral Borel resummation (49) we obtain the power series

$$\mathscr{S}_{\pm}\left[ \Phi_B^{(k)} \right](\xi, \Delta) \sim -(4\pi\xi)^{N-\tilde{\Delta}} \sum_{n=0}^{\infty} c_{B,n}^{(k)}(\Delta) \frac{\Gamma(n+1+\tilde{\Delta}-N)\sin(\pi\tilde{\Delta})}{\pi} (4\pi\xi)^{-n-1}, \tag{57}$$

where $\tilde{\Delta} = (2k+|B|+1)\Delta$. If we plug this expansion in (48) we obtain the transseries representation

$$\tilde{\zeta}_B(N, \xi, \Delta) = (-1)^{NB\,\theta(B)} \sum_{k=0}^{\infty} e^{-4\pi\xi k} e^{\pm i\pi k(k+|B|)\Delta} (4\pi\xi)^{N-\tilde{\Delta}} \left( \sum_{n=0}^{\infty} \frac{C_{B,n}^{(k)}(\Delta)}{(4\pi\xi)^{n+1}} \right), \tag{58}$$

where the perturbative coefficients $C_{B,n}^{(k)}(\Delta)$ in the $k$ instanton-anti-instanton background on top of the $B$-instanton topological sector are given by

$$C_{B,n}^{(k)}(\Delta) = -c_{B,n}^{(k)}(\Delta) \frac{\Gamma(n+1+\tilde{\Delta}-N)\sin(\pi\tilde{\Delta})}{\pi}. \tag{59}$$

These coefficients, as well as the $c_{B,n}^{(k)}(\Delta)$, are all real numbers whenever $\Delta \in \mathbb{R}$. The reason for this lies in how we rewrote the integrand (51) of the directional Laplace transform (49). In the transseries (48) we have factorised out the complex phase from the integrand, so that the function (51) appearing in the lateral Laplace transform (49) is manifestly real for $x \in [0, 1)$ and $\xi, \Delta \in \mathbb{R}$. However there is still a branch cut starting at $x = 1$ and that is why in (48) we need to take lateral Borel resummations where the factor $e^{\pm i\pi k(k+|B|)\Delta}$ coming from the discontinuity is just the transseries parameter.[10]

Once we have the expression (59) we note that for generic $\Delta$ the factor $\Gamma(n+1+\tilde{\Delta}-N)$ gives a factorial growth of the perturbative coefficients thus making the above expression (58) a purely formal object, i.e. a transseries representation. However as we send $\Delta \to 0$ we see that the $\sin(\pi\tilde{\Delta}) \to 0$ but $\Gamma(n+1+\tilde{\Delta}-N)$ develops a pole for every $n = 0, ..., N-1$, thus effectively truncating the expansion (57) to a degree $N-1$ polynomial in $\xi$ as already seen previously in the undeformed equation (11). For example if we take the $\Delta \to 0$ limit for the $\mathbb{CP}^1$ case $\sin(\pi\tilde{\Delta})\Gamma(n+1+\tilde{\Delta}-2)$ gives a finite non-zero contribution for $n = 0$ and $n = 1$ while vanishing for $n \geq 2$ so the transseries expansion (58) effectively reduces to

$$\tilde{\zeta}_B(2, \xi, 0) = \sum_{k=0}^{\infty} e^{-4\pi\xi k} (4\pi\xi)^2 \left( c_{B,0}^{(k)}(0)(4\pi\xi)^{-1} - c_{B,1}^{(k)}(0)(4\pi\xi)^{-2} \right)$$
$$= \sum_{k=0}^{\infty} e^{-4\pi\xi k} \frac{4\pi\xi + 2\psi(k+1) + 2\psi(k+|B|+1)}{[k!(k+|B|)!]^2} = \zeta_B(2, \xi) \tag{60}$$

---

[10] The sign $\pm$ of the phase is correlated with the direction of the lateral Laplace resummation as in (48). In here we use the same symbol to denote the formal transseries and its appropriate directional Borel-Ecalle resummation.

where we used the explicit form for the coefficients (56) to obtain precisely the same expression (16). One can easily check for different values of $N$ that the limit of the transseries (58) when $\Delta \to 0$ reproduces the same topological sector contribution $\zeta_B(N, \xi)$ written in equation (11) obtained from localization.

If we start from the very beginning with $\Delta = 0$, as we did in the supersymmetric case (4) leading to (11), we do not generate a transseries and there is no direct way to exploit resurgent analysis to extract non-perturbative information out of the purely perturbative, asymptotic power series. As a matter of fact there is not even an asymptotic power series to begin with since perturbation theory truncates after a finite number of loops due to the supersymmetric nature of the physical quantity under consideration. However, as soon as we break slightly supersymmetry by introducing this $\Delta$-deformation we immediately generate an infinite perturbative expansion, and in fact the full transseries, out of thin hair. As the Cheshire cat says [58]:

*"You may have noticed that I'm not all there myself."* .

Once we realise that for generic $\Delta$ we do indeed have a transseries we know from resurgent analysis that obviously the splitting of perturbative and non-perturbative part in (58) give rise to ambiguities as the directional Borel integral (50) is ill-defined for $\theta = 0$ since $\arg x = 0$ (and $\arg x = \pi$) is a Stokes direction for $\Phi_0^{(0)}(x)$. The branch cut begins at $x = 1$ and depending on how we dodge it, either from above or from below, we will generate non-perturbative "ambiguities" that are exactly compensated for by the non-perturbative terms in (58). We will promptly show that the resummation of the full transseries (58) give rise to an unambiguous result.

## 5.1 Cancellation of ambiguities

As just mentioned if instead of working with the full transseries (48)-(58) we were only to focus on the purely perturbative piece, i.e. the $k = 0$ term, in a given topological sector $B$, according to which resummation we decided to pick $\mathscr{S}_+\left[\Phi_B^{(0)}\right](\xi, \Delta)$ or $\mathscr{S}_-\left[\Phi_B^{(0)}\right](\xi, \Delta)$ we would find two different analytic continuations of the formal asymptotic expansion (57). Furthermore, even if the formal power series (57) is manifestly real for $\xi$ and $\Delta$ real, neither of the analytic continuation $\mathscr{S}_\pm\left[\Phi_B^{(0)}\right](\xi, \Delta)$ is, the difference between the two is purely imaginary and usually called an "ambiguity" in the resummation procedure. The presence of these "ambiguities" is due to the fact that we decided to split the full transseries (48) into perturbative and non-perturbative part. We can now show that if we consider the Borel-Ecalle resummation of the complete transseries (48)-(58), the ambiguities $(\mathscr{S}_+ - \mathscr{S}_-)\left[\Phi_B^{(k)}\right](\xi, \Delta)$ in each non-perturbative sector together with the jump in the transseries parameter $e^{\pm i\pi k(k+|B|)}$ precisely conspire to cancel out and give an unambiguous and real answer when $\xi$ and $\Delta$ are real.

To this end let us start with the ambiguity in the resummation of the purely perturbative

piece in the trivial topological sector $B = 0$:

$$(\mathcal{S}_+ - \mathcal{S}_-)\left[\Phi_0^{(0)}\right] = \int_0^\infty dx\, e^{-wx} x^{-N+\Delta}\left(\Phi_0^{(0)}(x + i\epsilon, \Delta) - \Phi_0^{(0)}(x - i\epsilon, \Delta)\right)$$

$$= \int_0^\infty dx\, e^{-wx} x^{-N+\Delta} \Phi_0^{(0)}(x + i\epsilon, \Delta)\left(1 - e^{2\pi i \Delta \lfloor x \rfloor(\lfloor x \rfloor + 2)}\right)$$

$$= \int_1^2 dx\, e^{-wx} x^{-N+\Delta} \Phi_0^{(0)}(x + i\epsilon, \Delta)\left(1 - e^{6\pi i \Delta}\right) +$$

$$+ \int_2^3 dx\, e^{-wx} x^{-N+\Delta} \Phi_0^{(0)}(x + i\epsilon, \Delta)\left(1 - e^{16\pi i \Delta}\right) + \mathrm{O}(e^{-3w}),$$

where we used the discontinuity equations (41-42) and defined $w = 4\pi\xi$. In each of the above integrals we can shift the integration variables to make manifest the exponentially suppressed factor, furthermore we also extend the integration all the way to infinity making sure that we are consistent with the order of the instanton counting parameter $e^{-w}$ at which we are working with. Proceeding as just outlined and using the connection formulas (52-53) we can rewrite the above equation as

$$(\mathcal{S}_+ - \mathcal{S}_-)\left[\Phi_0^{(0)}\right] = -2i\sin(\pi\Delta)e^{-w}\int_0^\infty dx\, e^{-wx} x^{-N+3\Delta} \Phi_0^{(1)}(x + i\epsilon, \Delta) + \tag{61}$$

$$- 2i\sin(\pi\Delta)e^{3i\pi\Delta}e^{-2w}\int_0^\infty dx\, e^{-wx} x^{-N+5\Delta} \Phi_0^{(2)}(x + i\epsilon, \Delta) + \mathrm{O}(e^{-3w})$$

$$= -2i\sin(\pi\Delta)e^{-w}\mathcal{S}_+\left[\Phi_0^{(1)}\right] - 2i\sin(\pi\Delta)e^{3i\pi\Delta}e^{-2w}\mathcal{S}_+\left[\Phi_0^{(2)}\right] + \mathrm{O}(e^{-3w}).$$

We were able to relate the difference between the two lateral resummations of the perturbative series to the resummation of the non-perturbative sectors, this relation is usually called *Stokes automorphism* (for more details see [11,12]). Note that the "ambiguity" in the resummation of the perturbative series is purely non-perturbative, i.e. it starts with $e^{-w}$ plus higher instantons sectors. This ambiguity does not look manifestly imaginary, however this is only due to the fact that the right-hand side is written in terms of the lateral resummation $\mathcal{S}_+\left[\Phi_0^{(k)}\right]$ of higher instanton sectors which is not a real quantity due to the branch cut running on the real axis for each $\Phi_0^{(k)}(x)$. We will obtain a manifest purely imaginary expression later on.

In a similar manner we can study what happens to the first non-perturbative sector, $k = 1$, in the transseries (48) and repeating a similar calculation we find:

$$e^{-w}\left(e^{i\pi\Delta}\mathcal{S}_+ - e^{-i\pi\Delta}\mathcal{S}_-\right)\left[\Phi_0^{(1)}\right] =$$

$$= +2i\sin(\pi\Delta)e^{-w}\mathcal{S}_+\left[\Phi_0^{(1)}\right] - 2i\sin(3\pi\Delta)e^{-i\pi\Delta}\mathcal{S}_+\left[\Phi_0^{(2)}\right] + \mathrm{O}(e^{-3w}). \tag{62}$$

Note that, unlike (61), the difference in lateral resummation of the $k = 1$ sector contains a term (the first one in the above expression) exactly of the same non-perturbative order $e^{-w}$. The reason for this is that we are not quite computing the ambiguity $(\mathcal{S}_+ - \mathcal{S}_-)\left[\Phi_0^{(1)}\right]$ but rather the joint combination of the jump in resummation together with the jump in the transseries parameter $e^{\pm i\pi\Delta}$.

Finally, to order $\mathrm{O}(e^{-3w})$ in the instanton counting parameter, we need to compute the "ambiguity" in the $k = 2$ non-perturbative sector of the transseries (48) given by

$$e^{-2w}\left(e^{4i\pi\Delta}\mathcal{S}_+ - e^{-4i\pi\Delta}\mathcal{S}_-\right)\left[\Phi_0^{(2)}\right] = +2i\sin(4\pi\Delta)e^{-2w}\mathcal{S}_+\left[\Phi_0^{(2)}\right] + \mathrm{O}(e^{-3w}). \tag{63}$$

Putting together the three pieces (61),(62), and (63) we obtain what expected

$$(\mathscr{S}_+ - \mathscr{S}_-)\left[\Phi_0^{(0)}\right] + e^{-w}\left(e^{i\pi\Delta}\mathscr{S}_+ - e^{-i\pi\Delta}\mathscr{S}_-\right)\left[\Phi_0^{(1)}\right] + e^{-2w}\left(e^{4i\pi\Delta}\mathscr{S}_+ - e^{-4i\pi\Delta}\mathscr{S}_-\right)\left[\Phi_0^{(2)}\right]$$
$$= O(e^{-3w}),$$

namely the difference in lateral resummation together with the correct jump in the transseries parameter combine and cancel out, giving a unique and unambiguous Borel-Ecalle resummation of the transseries (48).

From equations (61),(62), and (63) we can also rewrite the transseries (48) in a form which is manifestly real for real $\xi$ and $\Delta$ and absolutely unambiguous

$$\tilde{\zeta}_0(N,\xi,\Delta) = \mathscr{S}_0\left[\mathrm{Re}\left(\Phi_0^{(0)}\right)\right](\xi,\Delta) + \cos(\pi\Delta)e^{-4\pi\xi}\,\mathscr{S}_0\left[\mathrm{Re}\left(\Phi_0^{(1)}\right)\right](\xi,\Delta) +$$
$$+ \cos(\pi\Delta)\cos(3\pi\Delta)e^{-8\pi\xi}\,\mathscr{S}_0\left[\mathrm{Re}\left(\Phi_0^{(2)}\right)\right](\xi,\Delta) + O\left(e^{-12\pi\xi}\right). \tag{64}$$

Note that we do not need to take any lateral resummation now as the real part of the Borel transform $\mathrm{Re}\left(\Phi_0^{(0)}\right)$ does not have a branch cut along the positive real axis allowing us to safely perform the directional Borel transform $\mathscr{S}_0$ (50) without any ambiguity. We can repeat this analysis for generic topological sector $B$ obtaining a manifestly real and unambiguous resummation for the transseries (48)

$$\tilde{\zeta}_B(N,\xi,\Delta) = \mathscr{S}_0\left[\mathrm{Re}\left(\Phi_B^{(0)}\right)\right](\xi,\Delta) + \cos[(|B|+1)\pi\Delta]e^{-4\pi\xi}\,\mathscr{S}_0\left[\mathrm{Re}\left(\Phi_B^{(1)}\right)\right](\xi,\Delta) +$$
$$+ \cos[(|B|+1)\pi\Delta]\cos[(|B|+3)\pi\Delta]e^{-8\pi\xi}\,\mathscr{S}_0\left[\mathrm{Re}\left(\Phi_B^{(2)}\right)\right](\xi,\Delta) + O\left(e^{-12\pi\xi}\right). \tag{65}$$

## 5.2 Large orders relations

Now that we understand how the ambiguities in the resummation procedure cancel out when we consider the full transseries, we can try and use the purely perturbative data to retrieve some non-perturbative information. To proceed we consider $\Delta$ generic and use the transseries expansion (58) to extract the purely perturbative sector, now asymptotic, and only at the very end we will send $\Delta \to 0$ to learn something about the supersymmetric case. For simplicity let us focus on the perturbative part, $k = 0$, of the $B = 0$ topological sector in (58):

$$\tilde{\zeta}_{\mathrm{pert}}(N,\xi,\Delta) = \int_0^\infty dx\, e^{-4\pi\xi x}\, x^{-N+\Delta}\, \Phi_0^{(0)}(x,\Delta) \sim (4\pi\xi)^{N-\Delta}\sum_{n=0}^\infty \frac{C_{0,n}^{(0)}(\Delta)}{(4\pi\xi)^{n+1}}, \tag{66}$$

where the Borel transform obtained in (54) is

$$\Phi_0^{(0)}(x,\Delta) = -\frac{(-1)^N \sin(\pi\Delta)}{\pi}\left[\frac{\pi x/\sin(\pi x)}{\Gamma(1+x)^2}\right]^N \exp\left[2\Delta(x+\psi^{(-2)}(1))\right] \times$$
$$\exp\left[\Delta(2x+1)(\log\Gamma(1+x)-\log\Gamma(1-x)) - 2\Delta\left(\psi^{(-2)}(1+x)+\psi^{(-2)}(1-x)\right)\right],$$

and the perturbative coefficients (59)

$$C_{0,n}^{(0)}(\Delta) = -c_{0,n}^{(0)}(\Delta)\frac{\Gamma(n+1+\Delta-N)\,\sin(\pi\Delta)}{\pi} \tag{67}$$

can be obtained from (55) and grow factorially with $n$ for $\Delta$ generic.

Let us consider the particular determination of the resummation of the purely perturbative series (66), that we denote with the same symbol, where we anti-correlate $\arg\xi = \theta$ with the argument of the integration variable $x$ as:

$$(4\pi\xi)^{-N+\Delta}\tilde{\zeta}_{\mathrm{pert}}(N,\xi,\Delta) = \int_0^{\infty e^{-i\theta}} dx\, e^{-4\pi\xi x}(4\pi\xi x)^{-N+\Delta}\Phi_0^{(0)}(x,\Delta). \tag{68}$$

Note that this is not the correct physical quantity but rather it is the best we could do if we only had access to perturbation theory.

From the explicit expression (54) for $\Phi_0^{(0)}(x, \Delta)$ we know that the integrand of the above equation has two branch cuts along the Stokes directions $\arg x = 0$ starting at $x = +1$, and $\arg x = \pi$ starting at $x = -1$, thus forcing the determination for $\tilde{\zeta}_{\text{pert}}(N, \xi, \Delta)$ to have branch cuts along $\arg \xi = 0$ and $\arg \xi = \pi$. Using a standard Cauchy-like contour argument (see [6, 59]) we can relate the perturbative coefficients $C_{0,n}^{(0)}(\Delta)$ in the expansion (66), or more generically the one appearing in (58), to the discontinuities in the $\theta = 0$ and $\theta = \pi$ direction of the determination (68):

$$C_{0,n}^{(0)}(\Delta) \sim -\frac{1}{2\pi i} \int_0^\infty dw \, \text{Disc}_0(w) \, w^n - \frac{1}{2\pi i} \int_0^{\infty e^{i\pi}} dw \, \text{Disc}_\pi(w) w^n. \tag{69}$$

The discontinuities across the cuts of (68) can be easily computed using the discontinuity equations (41-42) for the $\log \Gamma$ and $\psi^{(-2)}$; in particular $\text{Disc}_0(w)$ vanishes for $0 < w < 1$ while $\text{Disc}_\pi(w)$ vanishes for $-1 < w < 0$. For example if we focus on

$$\text{Disc}_0(w) = \int_0^\infty dx \, e^{-wx} (wx)^{-N+\Delta} \left( \Phi_0^{(0)}(x - i\epsilon, \Delta) - \Phi_0^{(0)}(x + i\epsilon, \Delta) \right), \tag{70}$$

we can use multiple times (41-42) proceeding as we did in Section 5.1, and rewrite this expression as

$$\text{Disc}_0(w) = 2i \sin(\pi\Delta) e^{-w} w^{-2\Delta} \int_0^\infty dx \, e^{-wx} (wx)^{-N+3\Delta} \text{Re}\left( \Phi_0^{(1)}(x, \Delta) \right) \tag{71}$$

$$+ 2i \sin(\pi\Delta) \cos(3\pi\Delta) e^{-2w} w^{-4\Delta} \int_0^\infty dx \, e^{-wx} (wx)^{-N+5\Delta} \text{Re}\left( \Phi_0^{(2)}(x, \Delta) \right)$$

$$+ O\left( e^{-3w} \right).$$

One can also derive an expression for $\text{Disc}_\pi(w)$ and subsequently use equation (69) to obtain the asymptotic expansion valid at large $n \gg 1$ of the perturbative coefficients

$$C_{0,n}^{(0)}(\Delta) \sim -\frac{1}{2\pi} 2\sin(\pi\Delta) \frac{\Gamma(n-2\Delta)}{(+1)^{n-2\Delta}} \left( C_{0,0}^{(1)}(\Delta) + \frac{C_{0,1}^{(1)}(\Delta)}{n - 2\Delta - 1} + O(n^{-2}) \right) +$$

$$- \frac{1}{2\pi} 2\sin(\pi\Delta) \frac{\Gamma(n-4)}{(-1)^n} \left( C_{0,0}^{(-1)}(\Delta) + O(n^{-1}) \right) + \tag{72}$$

$$- \frac{1}{2\pi} 2\sin(\pi\Delta) \cos(3\pi\Delta) \frac{\Gamma(n-4\Delta)}{2^{n-4\Delta}} \left( C_{0,0}^{(2)}(\Delta) + \frac{2 C_{0,1}^{(2)}(\Delta)}{n - 4\Delta - 1} + O(n^{-2}) \right) + \dots .$$

From the large order perturbative coefficients coefficient $C_{0,n}^{(0)}$ we can disentangle the $C_{0,n}^{(k)}$ which are precisely the perturbative coefficients at order $n$ in the $k^{th}$ non-perturbative sector given in equation (59) and appearing in the transseries expansion (58). From perturbative data we can reconstruct non-perturbative physics. The second term in the asymptotic expansion would correspond to the $k = -1$ instanton-anti-instanton sector, i.e. something weighted by $e^{+4\pi\xi}$, and the first perturbative coefficient in this sector is given by

$$C_{0,0}^{(-1)}(\Delta) = \frac{\sin(\pi\Delta)}{\pi} (2\pi)^{-\Delta} \Gamma(3 + \Delta). \tag{73}$$

However we do not particularly care about these sectors as we specialised our transseries (58) to the wedge of the complex $\xi$ plane $\text{Re}\, \xi > 0$ and terms of the form $e^{+4\pi\xi}$ are unphysical here.

The large order perturbative coefficients do nonetheless know about these terms because if we were to analytically continue to the wedge $\operatorname{Re}\xi < 0$ terms of the form $e^{+4\pi\xi}$ would become exponentially suppressed and the most general transseries would contain both terms of the form $e^{\pm 4\pi k\xi}$. In particular, to be consistent, we should have written in (72) a term going as $\Gamma(n-\alpha)/(-2)^n$ however, as we will shortly see, in the supersymmetric limit $\Delta \to 0$ the $k < 0$ sectors will disappear completely as expected from the discussion in Section 2, while the footprints of the non-perturbative $k \geq 1$ sectors will still be present. The dots at the end of equation (72) represent all higher instanton sectors going as $\Gamma(n-\alpha_k)/(\pm k)^n$ for some constant $\alpha_k$, possibly $\Delta$ dependent.

We wrote equation (72) in a way that makes the Stokes constants for each non-perturbative sector manifest. For example for the $k = 1$ sector the Stokes constant is given by $A_1^{(0)} = 2\sin(\pi\Delta) = 2\operatorname{Im} e^{i\pi\Delta}$, i.e. the Stokes constant is exactly equal to the jump of the transseries parameter in the $k = 1$ instanton sector in equation (58) as expected since the Borel-Ecalle resummation of the transseries (48) should give the same result if we re-sum for $\arg\xi = +\epsilon$ or $\arg\xi = -\epsilon$. The Stokes constant for the $k = 2$ sector is however $A_2^{(0)} = 2\sin(\pi\Delta)\cos(3\pi\Delta)$ and does not equal the jump of $2\operatorname{Im} e^{4i\pi\Delta}$ in the transseries parameter for the $k = 2$ sector in (58). The reason is that the jump in the two instanton sector is compensated partly from the term $e^{-2w}$ in the discontinuity in the $k = 0$ sector in (71) but also from a term $e^{-w}$ in the discontinuity for the $k = 1$ sector, see (62). It is only the sum of these two pieces that reproduces the jump of $2\operatorname{Im} e^{4i\pi\Delta}$ in the transseries parameter for the $k = 2$ sector. To show that this is indeed the case we can first easily repeat the large order analysis for the perturbative coefficients $C_{0,n}^{(1)}$ in the $k = 1$ non-perturbative sector obtaining

$$C_{0,n}^{(1)}(\Delta) \sim -\frac{1}{2\pi}2\sin(3\pi\Delta)\frac{\Gamma(n+2\Delta)}{(-1)^n}\left(C_{0,0}^{(0)}(\Delta) + \frac{(-1)\,C_{0,1}^{(0)}(\Delta)}{n+2\Delta-1} + \mathrm{O}(n^{-2})\right) +$$

$$-\frac{1}{2\pi}2\sin(3\pi\Delta)\frac{\Gamma(n)}{(+1)^n}\left(C_{0,0}^{(2)}(\Delta) + \frac{C_{0,1}^{(2)}(\Delta)}{n-1} + \mathrm{O}(n^{-2})\right) + \dots, \qquad (74)$$

where the dots represent higher non-perturbative contributions as above. The $k = 1$ sector "sees" the perturbative sector with a relative action of $-1$ hence the alternating factor $(-1)^n$ in the first term multiplying exactly the purely perturbative coefficients $C_{0,n}^{(0)}$ with Stokes constant $A_{-1}^{(1)} = 2\sin(3\pi\Delta)$. The relative action between the $k = 1$ sector and the $k = 2$ sector is instead equal to $+1$ hence the second term in the above equation does not have an alternating factor and multiplies the perturbative coefficients $C_{0,n}^{(2)}$ of the $k = 2$ sector with Stokes constant $A_1^{(1)} = 2\sin(3\pi\Delta)$. We can now see that the jump $2\operatorname{Im} e^{4i\pi\Delta}$ of the $k = 2$ transseries parameter in (58) is exactly controlled by the Stokes constant $A_2^{(0)} = 2\sin(\pi\Delta)\cos(3\pi\Delta)$ of the perturbative sector plus the Stokes constant $A_1^{(1)} = 2\sin(3\pi\Delta)$ of the $k = 1$ sector multiplied by the real part $\operatorname{Re} e^{i\pi\Delta}$ of the transseries parameter[11] for the $k = 1$ sector

$$2\operatorname{Im} e^{4i\pi\Delta} = A_2^{(0)} + A_1^{(1)}\operatorname{Re} e^{i\pi\Delta} = 2\sin(\pi\Delta)\cos(3\pi\Delta) + 2\sin(3\pi\Delta)\cos(\pi\Delta) = 2\sin(4\pi\Delta).$$

The large order coefficients (72-74) are a genuine factorial asymptotic expansion for generic $\Delta$. As a numerical check we can fix $\Delta$ to some value and read from the large order perturbative coefficients (72) the low order non-perturbative sector coefficients. A curious incident happens whenever we pick a rational $\Delta = p/q$ for some coprime integers $p, q \in \mathbb{Z}$.

---

[11]In (48) one considers the jump of the $k = 1$ sector $e^{i\pi\Delta}\mathscr{S}_+[\Phi_0^{(1)}] - e^{-i\pi\Delta}\mathscr{S}_-[\Phi_0^{(1)}]$ and the only term in this expression contributing to the $k = 2$ sector is given by $\operatorname{Re}\left(e^{i\pi\Delta}\right) \times (\mathscr{S}_+ - \mathscr{S}_-)[\Phi_0^{(1)}] \sim 2i\operatorname{Re}\left(e^{i\pi\Delta}\right)A_1^{(1)}e^{-8\pi\xi}$. See the thorough discussion in Section 5.1 and in particular equation (62).

From equation (55) we see that in all the instanton sectors where $(2k + |B| + 1) = 0\,(\mathrm{mod}\,q)$ due to the $\sin((2k + |B| + 1)\pi\Delta)$ factor we have a truncation and the perturbative coefficients $C_{B,n}^{(k)}$ in those non-perturbative sectors are not asymptotic but rather finite in number. In all the sectors for which $(2k + |B| + 1) \neq 0\,(\mathrm{mod}\,q)$, in particular the purely perturbative one, the coefficients remain asymptotic and this truncation seems of accidental nature. However as we will comment later on in Section 5.3 whenever $\Delta \in \mathbb{Z}$ we have that this truncation happens in *all* sectors giving rise to some "exact" observable, as in the case $\Delta = 0$ discussed in detail above. As a nice example of this accidental truncation we can pick the large order expansion (72) and specialise it to the case $\Delta = 1/3$. Fixing for concreteness $N = 2$, i.e. $\mathbb{CP}^1$, and for the particular value $\Delta = 1/3$, we have that the $k = 1$ sector truncates dramatically

$$C_{0,0}^{(1)}(1/3) = \sqrt[3]{\frac{e^4}{2\pi}}\,,$$

$$C_{0,n}^{(1)}(1/3) = 0\,, \qquad n \geq 1\,,$$

note that for larger $N$ these coefficients would truncate after $N - 1$ orders. Using (72) we obtain the asymptotic form of the perturbative coefficients

$$C_{0,n}^{(0),\mathrm{as}}(1/3) = -\frac{\sin(\pi/3)}{\pi}\Gamma(n - 2/3)\,C_{0,0}^{(1)}(1/3) \tag{75}$$

using (73) for $\Delta = 1/3$, so according to (72) for $n \gg 1$ the difference between the perturbative coefficients and (75) will tell us about the first sub-leading correction:

$$\left[\frac{-\pi\left(C_{0,n}^{(0)}(1/3) - C_{0,n}^{(0),\mathrm{as}}(1/3)\right)}{\sin(\pi/3)\Gamma(n - 2/3)}\right] \sim \frac{(-1)^n}{n^{10/3}}C_{0,0}^{(-1)}(1/3) + \mathrm{O}(n^{-13/3})$$

$$\sim \frac{(-1)^n}{n^{10/3}}\frac{\sin(\pi/3)}{\pi}(2\pi)^{-1/3}\Gamma(10/3)\,.$$

In Figure 3 we plot the difference between the perturbative coefficients $C_{0,n}^{(0)}(1/3)$ computed via (59) and their asymptotic form $C_{0,n}^{(0),\mathrm{as}}(1/3)$ just presented in (75):

$$d_n = \left[\frac{-\pi\left(C_{0,n}^{(0)}(1/3) - C_{0,n}^{(0),\mathrm{as}}(1/3)\right)}{\sin(\pi/3)\Gamma(n - 2/3)}\right](-1)^n n^{10/3} \tag{76}$$

$$\sim \frac{\sin(\pi/3)}{\pi}(2\pi)^{-1/3}\Gamma(10/3) + \mathrm{O}(n^{-1})\,. \tag{77}$$

For a generic value of $\Delta$ we can read the non-perturbative coefficients from the large order perturbative ones.

We want to understand now what happens to the asymptotic forms (72-74) when $\Delta \to 0$, i.e. when we reach the supersymmetric point. As we already saw below equation (59), when we send $\Delta \to 0$ in every non-perturbative sector only the first two perturbative coefficients $C_{0,0}^{(k)}(0)$ and $C_{0,1}^{(k)}(0)$ survive, while all the others vanish. It would seem that there is no way to reconstruct from the perturbative coefficients some non-perturbative physics and vice-versa because the asymptotic forms (72-74) do not hold; the series are not asymptotic but they drastically truncate. However the footprints of the Cheshire cat resurgence are still there! If we consider the asymptotic form (72) but rather study the coefficients $-c_{0,n}^{(0)}(\Delta)$ using (67) we

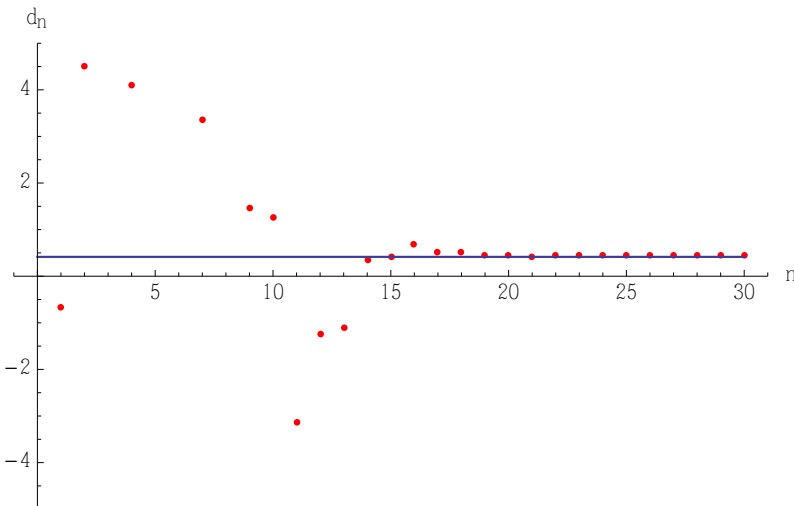

Figure 3: Difference $d_n$ between the perturbative coefficients $C_{0,n}^{(0)}(1/3)$ and their asymptotic form $C_{0,n}^{(0),\text{as}}(1/3)$. The blue line is given by the equation $y = C_{0,0}^{(-1)}(1/3) = \frac{\sin(\pi/3)}{\pi}(2\pi)^{-1/3}\Gamma(10/3) \simeq 0.415$.

have

$$c_{0,n}^{(0)}(\Delta) = \frac{-\pi\, C_{0,n}^{(0)}(\Delta)}{\sin(\pi\Delta)\,\Gamma(n+1+\Delta-N)}$$

$$\sim \frac{\Gamma(n-2\Delta)}{\Gamma(n+1+\Delta-N)(+1)^{n-2\Delta}}\left(C_{0,0}^{(1)}(\Delta) + \frac{C_{0,1}^{(1)}(\Delta)}{n-2\Delta-1} + \mathrm{O}(n^{-2})\right) +$$

$$+ \frac{\Gamma(n-4)}{\Gamma(n+1+\Delta-N)(-1)^n}\left(C_{0,0}^{(-1)}(\Delta) + \mathrm{O}(n^{-1})\right) +$$

$$+ \cos(3\pi\Delta)\frac{\Gamma(n-4\Delta)}{\Gamma(n+1+\Delta-N)\,2^{n-4\Delta}}\left(C_{0,0}^{(2)}(\Delta) + \frac{2\,C_{0,1}^{(2)}(\Delta)}{n-4\Delta-1} + \mathrm{O}(n^{-2})\right). \quad (78)$$

We can now safely send $\Delta \to 0$ and the coefficients $c_{0,n}^{(0)}(0)$ will not truncate. As we set $\Delta = 0$ in the right-hand side the first thing to notice is that the contributions of the form $\Gamma(n-\alpha_k)/(-k)^n$, corresponding to the presence of exponentially enhanced terms $e^{+4\pi k\xi}$ in the transseries, all disappear since all the coefficients $C_{0,n}^{(-k)}(0) = 0$ when $k > 0$, see equation (73). This is expected since in the supersymmetric case $\Delta = 0$ these terms were not present in (16). Furthermore on physical grounds we do not expect terms exponentially enhanced to appear in the expansion of any physical quantity. On the other hand for $k \in \mathbb{N}$ we know that the perturbative coefficients $C_{0,n}^{(k)}(0)$ in the $\mathbb{CP}^{N-1}$ model are non-vanishing only for $n \leq N-1$. For concreteness in the $\mathbb{CP}^1$ case in each non-perturbative sector only the first two perturbative terms are non-vanishing as we already saw in equation (16), and the asymptotic expansion (78) reduces to

$$c_{0,n}^{(0)}(0) \sim \frac{(n-1)}{1^n}\left(C_{0,0}^{(1)}(0) + \frac{C_{0,1}^{(1)}(0)}{n-1}\right) + \frac{(n-1)}{2^n}\left(C_{0,0}^{(2)}(0) + \frac{2\,C_{0,1}^{(2)}(0)}{n-1}\right) + \mathrm{O}(n\,3^{-n}), \quad (79)$$

where the non-perturbative coefficients $C_{0,0}^{(k)}(0)$, and $C_{0,1}^{(k)}(0)$ can be obtained from (59) and

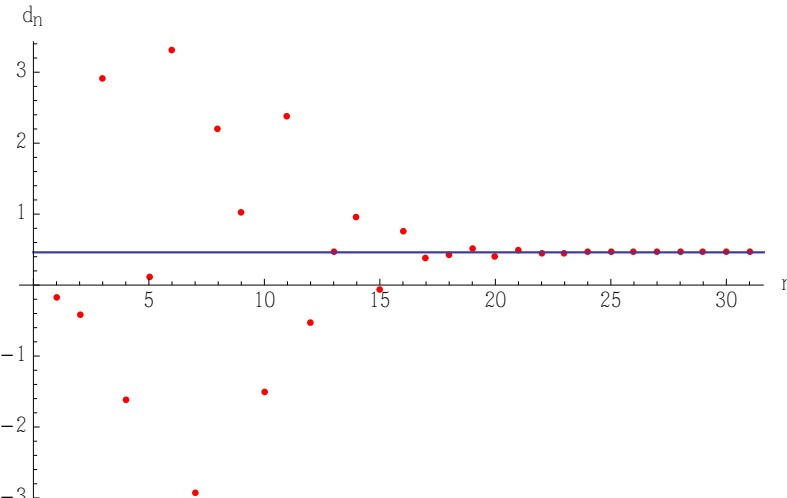

Figure 4: Difference $d_n$ between the perturbative coefficients $c_{0,n}^{(0)}(0)$ and their asymptotic form $c_{0,n}^{(0),\text{as}}(0)$. The blue line is given by the equation $y = 2\,C_{0,1}^{(2)}(0) = \frac{1}{4}(3-4\gamma) \simeq 0.461$.

(56) and reproduce precisely the coefficients in the supersymmetric expansion (17)

$$C_{0,0}^{(k)}(0) = \frac{1}{(k!)^4}, \qquad\qquad C_{0,1}^{(k)}(0) = \frac{4H_k - 4\gamma}{(k!)^4}. \qquad (80)$$

Note that equation (79) is actually not an asymptotic expansion and could have been derived in the supersymmetric case by considering the undeformed integrand of (4), writing the coefficient $c_{0,n}^{(0)}(0)$ as a Cauchy integral around the origin and then closing the contour so to get the contribution from all the other poles, i.e. the non-perturbative sectors. This is of course possible because the partition function (4) does contain all the information, perturbative and non-perturbative. However had we been given *only* the perturbative coefficients in the supersymmetric case it would have been impossible to reconstruct the non-perturbative data without the aid of Cheshire cat resurgence.

As a numerical check we can define the asymptotic approximation

$$c_{0,n}^{(0),\,\text{as}}(0) = \frac{(n-1)}{(+1)^n}\left(1 + \frac{4-4\gamma}{n-1}\right) + \frac{(n-1)}{(+2)^n}\frac{1}{16}, \qquad (81)$$

where we made explicit use of (80). From the difference between the perturbative coefficients $c_{0,n}^{(0)}$, that we can easily generate from (54) and (55), and the asymptotic form (81) we can extract non-perturbative information out of perturbative data

$$d_n = \left(c_{0,n}^{(0)}(0) - c_{0,n}^{(0),\,\text{as}}(0)\right)2^n \sim 2C_{0,1}^{(2)}(0) + \text{O}\left((2/3)^n\right) \sim \frac{1}{4}(3-4\gamma). \qquad (82)$$

In Figure 4 we show how this difference $d_n$ tends to $2C_{0,1}^{(2)}(0)$ allowing us to reconstruct the perturbative coefficients of the non-perturbative sectors. Surprisingly enough it is still possible to extract non-perturbative data from perturbation theory even when the perturbative expansion truncates: the Cheshire cat's grin still lingers on even when his body has completely disappeared.

We can repeat this story also for the large order form of the non-perturbative sectors coef-

ficients. We can consider the $k = 1$ sector and rewrite equation (74) using (59)

$$
\begin{aligned}
c_{0,n}^{(1)}(0) &= \frac{-\pi\, C_{0,n}^{(1)}(\Delta)}{\sin(3\pi\Delta)\,\Gamma(n+1+3\Delta-N)} \\
&\sim \frac{\Gamma(n+2\Delta)}{\Gamma(n+1+3\Delta-N)(-1)^n}\left(C_{0,0}^{(0)}(\Delta) + \frac{(-1)\,C_{0,1}^{(0)}(\Delta)}{n+2\Delta-1} + \mathrm{O}(n^{-2})\right) + \\
&\quad + \frac{\Gamma(n)}{\Gamma(n+1+3\Delta-N)(+1)^n}\left(C_{0,0}^{(2)}(\Delta) + \frac{C_{0,1}^{(2)}(\Delta)}{n-1} + \mathrm{O}(n^{-2})\right) + \dots.
\end{aligned}
\tag{83}
$$

For concreteness we fix once more $N = 2$, i.e. $\mathbb{CP}^1$, so that when we take the limit $\Delta \to 0$ we have only two non-vanishing perturbative coefficients in each sector and in this limit the above equation becomes

$$
c_{0,n}^{(1)}(0) \sim \frac{(n-1)}{(-1)^n}\left(C_{0,0}^{(0)}(0) - \frac{C_{0,1}^{(0)}(0)}{n-1}\right) + \frac{(n-1)}{1^n}\left(C_{0,0}^{(2)}(0) + \frac{C_{0,1}^{(2)}(0)}{n-1}\right) + \mathrm{O}(n\,2^{-n}).
$$

Since the $k = 1$ sector "sees" the perturbative sector with a relative action of $-1$, while the $k = 2$ sector with a relative action of $+1$, we have two competing saddles here and find an oscillating behaviour. We can define the asymptotic approximation

$$
c_{0,n}^{(1),\,\mathrm{as}}(0) = \frac{(n-1)}{(-1)^n} + \frac{(n-1)}{(+1)^n}\left(\frac{1}{16} - \frac{3-4\gamma}{8(n-1)}\right),
\tag{84}
$$

where we made explicit use of (80). If we consider the difference between the perturbative coefficients in the $k = 1$ non-perturbative sector, easily obtained from (51-55), and the asymptotic approximation just defined we have

$$
d_n = \left(c_{0,n}^{(1)}(0) - c_{0,n}^{(1),\,\mathrm{as}}(0)\right)(-1)^n \sim -C_{0,1}^{(0)}(0) + \mathrm{O}(2^{-n}) \sim 4\gamma,
\tag{85}
$$

and in Figure 5 we see how we can reconstruct the purely perturbative coefficients out of the perturbative data in a given non-perturbative sector even when all the perturbative expansions truncate to a finite number of terms.

## 5.3 Other solvable observables

So far we have considered in detail only the limit $\Delta \to 0$ for which the body of the Cheshire cat disappears and we find once more the convergent supersymmetric result. However from equation (59) we can see that more generically we just need $\Delta$ to approach an integer and *all* the topological sectors perturbative expansions will truncate after a finite number of terms. For example in the perturbative sector $k = 0, B = 0$, we read from (59) that whenever $\Delta \to n \in \mathbb{N}$ the perturbative coefficients truncate after $N - n - 1$ orders, so fewer orders than the $\Delta \to 0$ case. From (59) we see that in higher topological number sectors we obtain even fewer perturbative coefficients. It is suggestive to go back to our modified one-loop determinant (38) and reinterpret this truncation when $\Delta = N_f - N_b \to n \in \mathbb{N}$ as perhaps the insertion of some supersymmetric fermionic operator.

Similarly when $\Delta$ approaches a negative integer, $-\Delta = m \in \mathbb{N}$, the perturbative coefficients in the $k = 0, B = 0$, truncate after $N + m - 1$ orders hence we find more coefficients than the $\Delta = 0$ case. Contrary to before we can see from (59) that in higher and higher topological number sectors we obtain more and more perturbative coefficients. Again this increase in perturbative coefficients can be seen from the modified one-loop determinant (38) interpreting the limit $-\Delta = N_b - N_f \to m \in \mathbb{N}$ as the insertion of some supersymmetric bosonic operator.

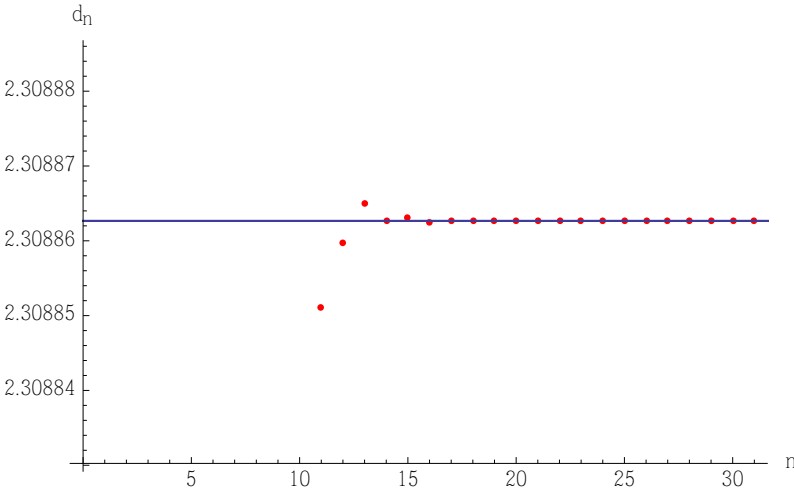

Figure 5: Difference $d_n$ between the perturbative coefficients $c_{0,n}^{(1)}(0)$ and their asymptotic form $c_{0,n}^{(1),\text{as}}(0)$. The blue line is given by the equation $y = -C_{0,1}^{(0)}(0) = 4\gamma \simeq 2.308$.

It would be tempting to interpret these results as the genuine modification of the original path integral with an unequal (but integer) number of bosons and fermions. However we should stress once more that our modification to the one-loop determinant (38) effectively takes place only after having heavily exploited the supersymmetry of the model to localize the path integral. It is nonetheless striking to notice the similarity of our truncation of the perturbative coefficients when $\Delta \to n \in \mathbb{Z}$ with the quasi-solvability discussed in [34]. As mentioned in the Introduction the authors of [34] consider an analytic continuation in the number of fermions $\zeta$ and they found that in the double Sine-Gordon quantum mechanics the lowest $\zeta$ states are algebraically solvable when $\zeta \in \mathbb{N}$ and the exact energies of these levels can be exactly computed and are algebraic functions of the coupling constant.

## 6 Resurgence from analytic continuation in $N$

An alternative way to obtain Cheshire cat resurgence for the $\mathbb{CP}^{N-1}$ model is to turn off the supersymmetry breaking deformation $\Delta$ and instead consider an analytic continuation in the number of chiral multiplets from $N \in \mathbb{N}$ to $r \in \mathbb{R}$ (or $\mathbb{C}$) thus studying the undeformed partition function (4) but for a $\mathbb{CP}^{r-1}$ model (one can also consider both deformations at once). Unlike the previously discussed case $\Delta \neq 0$, this deformation is of a more supersymmetric nature and the supersymmetry algebra is still formally unchanged and satisfied. Nonetheless for generic $r \in \mathbb{R}$ we will show that perturbation theory is asymptotic and truncates precisely when $r \to N \in \mathbb{N}$.

When $N$ is replaced by $r \in \mathbb{R}$ the poles and zeroes of the original partition function become branch cuts for the undeformed one-loop determinant and for $r > 0$ (or $\text{Re}\, r > 0$) we can write the partition function as we did in (43)

$$Z(r) = \sum_{B \in \mathbb{Z}} e^{-i\theta B} \int_{\mathscr{C}} \frac{d\sigma}{2\pi} e^{-4\pi i \xi \sigma} \tilde{Z}_{matter}(\sigma), \tag{86}$$

where the deformed one-loop determinant can be obtained from (40) after having set $\Delta = 0$

$$\tilde{Z}_{matter}(\sigma) = e^{i\pi B \theta(B) r} \exp\left[ r \left( \log \Gamma(-i\sigma + |B|/2) - \log \Gamma(1 + i\sigma + |B|/2) \right) \right], \tag{87}$$

which reproduces the original supersymmetric result whenever $r = N \in \mathbb{N}$. As previously discussed the contour of integration $\mathscr{C}$ comes from $\sigma \to -i\infty - \epsilon$, circles around the origin and then goes back to $\sigma \to -i\infty + \epsilon$, for $r < 0$ (or $\mathrm{Re}\, r < 0$) we simply close the contour around the positive imaginary axis.

At this point we can repeat the same procedure we followed in Section 4, realising that the discontinuity in (87) now comes only from the $\log \Gamma$ function and, after using the discontinuity property (41), we obtain

$$Z(r) = \sum_{B \in \mathbb{Z}} e^{-2\pi\xi|B| - i\theta B} \, \tilde{\zeta}_B(r, \xi), \tag{88}$$

where each Fourier mode can be written as

$$
\begin{aligned}
\tilde{\zeta}_B(r, \xi) &= \sum_{k=0}^{\infty} \int_k^{k+1} \frac{dx}{2\pi i} e^{-4\pi\xi x} \tilde{Z}_{matter}(-ix - i|B|/2 - \epsilon)\left[e^{2\pi ikr} - 1\right] \\
&= \sum_{k=0}^{\infty} \int_k^{k+1} \frac{dx}{2\pi i} e^{-4\pi\xi x} \tilde{Z}_{matter}(-ix - i|B|/2 + \epsilon)\left[1 - e^{-2\pi ikr}\right].
\end{aligned}
\tag{89}
$$

Similarly to what we did before we rewrite $\int_k^{k+1} = \int_k^{\infty} - \int_{k+1}^{\infty}$ and shift variables so that every integral becomes between $[0, \infty)$ arriving at the transseries expansion

$$\tilde{\zeta}_B(r, \xi) = e^{i\pi B\theta(B)r} \sum_{k=0}^{\infty} e^{-4\pi\xi k} e^{\mp i\pi kr} \, \tilde{\mathscr{S}}_{\pm}\left[\tilde{\Phi}_B^{(k)}\right](\xi, r), \tag{90}$$

where $\tilde{\mathscr{S}}_{\pm}$ denote the modified lateral Laplace transforms

$$\tilde{\mathscr{S}}_{\pm}\left[\tilde{\Phi}_B^{(k)}\right](\xi, r) = \int_0^{\infty \pm i\epsilon} dx \, e^{-4\pi\xi x} x^{-r} \tilde{\Phi}_B^{(k)}(x, r), \tag{91}$$

and, after repeated use of the connection formula (52), the Borel transform $\tilde{\Phi}_B^{(k)}(x, r)$ is given by

$$\tilde{\Phi}_B^{(k)}(x, r) = \frac{\sin(\pi r)}{\pi} \exp\left[r\left(\log\Gamma(1-x) - \log\Gamma(x+k+|B|+1) - \sum_{j=1}^{k} \log(x+j)\right)\right]. \tag{92}$$

Comparing these equations to (48)-(49)-(51) obtained in Section 4, we see that the role played by the deformation parameter $\Delta$ is now taken by $r$. If we expand the Borel transform for $x \sim 0$ we get

$$\tilde{\Phi}_B^{(k)}(x, r) \sim \frac{\sin(\pi r)}{\pi} \left(\sum_{n=0}^{\infty} \tilde{c}_{B,n}^{(k)}(r) x^n\right). \tag{93}$$

We see that in the limit $r \to N \in \mathbb{N}$ we obtain precisely the same coefficients (56) previously found in the limit $\Delta \to 0$.

So if we consider a weak coupling expansion, i.e. $\xi \to \infty$, of the lateral Borel resummation (91) we obtain the power series

$$\tilde{\mathscr{S}}_{\pm}\left[\tilde{\Phi}_B^{(k)}\right](\xi, r) \sim (4\pi\xi)^r \sum_{n=0}^{\infty} \tilde{c}_{B,n}^{(k)}(r) \frac{\Gamma(n+1-r)\sin(\pi r)}{\pi} (4\pi\xi)^{-n-1}. \tag{94}$$

If we plug this expansion in (90) we obtain the transseries representation

$$\tilde{\zeta}_B(r, \xi) = e^{i\pi B\theta(B)r} \sum_{k=0}^{\infty} e^{-4\pi\xi k} e^{\mp i\pi kr} (4\pi\xi)^r \left(\sum_{n=0}^{\infty} \frac{\tilde{C}_{B,n}^{(k)}(r)}{(4\pi\xi)^{n+1}}\right), \tag{95}$$

where the perturbative coefficients $\tilde{C}_{B,n}^{(k)}(r)$ in the $k$ instanton-anti-instanton background on top of the $B$-instanton topological sector are given by

$$\tilde{C}_{B,n}^{(k)}(r) = \tilde{c}_{B,n}^{(k)}(r)\frac{\Gamma(n+1-N)\,\sin(\pi r)}{\pi}, \tag{96}$$

and the sign of the transseries parameter $e^{\mp i\pi kr}$ is correlated with the direction of the Lateral resummation as in (90).

These coefficients (96) are, for generic $r \in \mathbb{R}$, factorially diverging and the above expression (95) is a purely formal object, i.e. a transseries representation. However as we send $r \to N \in \mathbb{N}$ we see that the $\sin(\pi r) \to 0$ but $\Gamma(n+1-r)$ develops a pole for every $n = 0, ..., N-1$, thus effectively truncating the expansion (94) to a degree $N-1$ polynomial in $\xi$ reproducing the undeformed equation (11) for $\mathbb{CP}^{N-1}$, in an identical fashion to the limit $\Delta \to 0$ for deformed case (57). Although formally still supersymmetric, the $\mathbb{CP}^{r-1}$ model with $r \in \mathbb{R}$ produces asymptotic perturbative expansions, truncating only in the limit $r \to N \in \mathbb{N}$.

## 6.1 Cancellation of ambiguities

Using the formulas just derived we can repeat also in the present $\mathbb{CP}^{r-1}$, $r \in \mathbb{R}$, case the same analysis carried out in Section 5 for the $\Delta$ deformed model. In particular we can show that the ambiguities in resummation cancel out in (90) and that the discontinuity for the resummation of the purely perturbative sector contains all the non-perturbative data. To this end we can analyse the difference in lateral resummations, i.e. the Stokes automorphism,

$$(\tilde{\mathscr{S}}_+ - \tilde{\mathscr{S}}_-)\left[\tilde{\Phi}_B^{(k)}\right] = \int_0^\infty dx\, e^{-wx} x^{-r}\left(\tilde{\Phi}_B^{(k)}(x+i\epsilon, r) - \tilde{\Phi}_B^{(k)}(x-i\epsilon, r)\right)$$

$$= 2i\sin(\pi r)\sum_{n=1}^\infty e^{-nw} e^{\mp i\pi(n-1)r}\tilde{\mathscr{S}}_\pm\left[\tilde{\Phi}_B^{(k+n)}\right] \tag{97}$$

where we made intensive use of the discontinuity property (41) and connection formula (52) for the $\log\Gamma$ function and denoted $4\pi\xi = w$.

We can now prove that all the ambiguities cancel out in (90) by considering the difference between the two lateral resummations together with the jump in the transseries parameter:

$$\sum_{k=0}^\infty e^{-kw}\left(e^{-i\pi kr}\tilde{\mathscr{S}}_+ - e^{i\pi kr}\tilde{\mathscr{S}}_-\right)\left[\tilde{\Phi}_B^{(k)}\right]$$

$$= \sum_{k=0}^\infty -2i\sin(\pi kr)e^{-kw}\tilde{\mathscr{S}}_+\left[\tilde{\Phi}_B^{(k)}\right] + \sum_{k=0}^\infty e^{i\pi kr}e^{-kw}(\tilde{\mathscr{S}}_+ - \tilde{\mathscr{S}}_-)\left[\tilde{\Phi}_B^{(k)}\right]$$

$$= \sum_{k=0}^\infty -2i\sin(\pi kr)e^{-kw}\tilde{\mathscr{S}}_+\left[\tilde{\Phi}_B^{(k)}\right] + \sum_{k=0}^\infty\sum_{n=1}^\infty 2i\sin(\pi r)e^{-(k+n)w}e^{i\pi(k-n+1)r}\tilde{\mathscr{S}}_+\left[\tilde{\Phi}_B^{(k+n)}\right]$$

$$= \sum_{k=0}^\infty -2i\sin(\pi kr)e^{-kw}\tilde{\mathscr{S}}_+\left[\tilde{\Phi}_B^{(k)}\right] + \sum_{m=1}^\infty 2i\sin(\pi r)e^{-mw}\tilde{\mathscr{S}}_+\left[\tilde{\Phi}_B^{(m)}\right]\sum_{k=0}^\infty\sum_{n=1}^\infty\delta_{k+n,m}e^{i\pi(k-n+1)r}$$

$$= \sum_{k=0}^\infty -2i\sin(\pi kr)e^{-kw}\tilde{\mathscr{S}}_+\left[\tilde{\Phi}_B^{(k)}\right] + \sum_{m=1}^\infty 2i\sin(\pi r)\frac{\sin(\pi mr)}{\sin(\pi r)}e^{-mw}\tilde{\mathscr{S}}_+\left[\tilde{\Phi}_B^{(m)}\right] = 0,$$

where we made use of the Stokes automorphism (97).

Similarly to Section 5, see equation (68), we can also define the analytic continuation obtained from the purely perturbative coefficients

$$(4\pi\xi)^{-r}\tilde{\zeta}_{\text{pert}}(r, \xi) = \int_0^{\infty e^{-i\theta}} dx\, e^{-4\pi\xi x}(4\pi\xi x)^{-r}\tilde{\Phi}_0^{(0)}(x, r), \tag{98}$$

with $\theta = \arg \xi$. From equation (92) we deduce that this function has two branch cuts along the complex directions $\arg \xi = 0$ and $\pi$, and it is a matter of simple calculations to show that its discontinuity across the real positive axis is given by

$$
\begin{aligned}
\mathrm{Disc}_0(w) &= \int_0^\infty dx \, e^{-wx} (wx)^{-r} \left( \tilde{\Phi}_0^{(0)}(x - i\epsilon, r) - \tilde{\Phi}_0^{(0)}(x + i\epsilon, r) \right) \\
&= -2i \sin(\pi r) e^{-w} \int_0^\infty dx \, e^{-wx} (wx)^{-r} \, \mathrm{Re}\left( \tilde{\Phi}_0^{(1)}(x, r) \right) \\
&\quad - i \sin(2\pi r) e^{-2w} \int_0^\infty dx \, e^{-wx} (wx)^{-r} \, \mathrm{Re}\left( \tilde{\Phi}_0^{(2)}(x, r) \right) + \mathrm{O}\left( e^{-3w} \right),
\end{aligned} \tag{99}
$$

similar to what we obtained in the $\Delta$ deformed case (71). An analog equation can be obtained for the discontinuity across the negative real axis.

From the discontinuity we can read the Stokes constants and as expected the Stokes constant $\tilde{A}_1^{(0)} = -2\sin(\pi r)$ is exactly equal to the jump $2\,\mathrm{Im}\, e^{-i\pi r}$ of the transseries parameter in the $k = 1$ instanton sector in equation (90). Furthermore, similarly to the $\Delta$ deformed case, the Stokes constant $\tilde{A}_2^{(0)} = -\sin(2\pi r)$ for the $k = 2$ sector does not equal the jump $2\,\mathrm{Im}\, e^{-2i\pi r}$ in the transseries parameter for the $k = 2$ sector in (95). The reason is of course that the jump in the two instanton sector is compensated partly from the term $e^{-2w}$ in the discontinuity for the $k = 0$ sector in (99) but also from a term $e^{-w}$ in the discontinuity for the $k = 1$ sector that can be similarly computed and produces a Stokes constant $\tilde{A}_1^{(1)} = -2\sin(\pi r)$. The jump $2\,\mathrm{Im}\, e^{-2i\pi r}$ of the $k = 2$ transseries parameter in (95) is exactly controlled by the Stokes constant $\tilde{A}_2^{(0)}$ of the perturbative sector plus the Stokes constant $\tilde{A}_1^{(1)}$ of the $k = 1$ sector multiplied by the real part $\mathrm{Re}\, e^{-i\pi r}$ of the transseries parameter for the $k = 1$ sector

$$
2\,\mathrm{Im}\, e^{-2i\pi r} = \tilde{A}_2^{(0)} + \tilde{A}_1^{(1)} \, \mathrm{Re}\, e^{-i\pi r} = -\sin(2\pi r) - 2\sin(\pi r)\cos(\pi r) = -2\sin(2\pi r).
$$

From the above discussion it is a simple exercise to obtain the large order behaviour of the perturbative coefficients, as we did in Section 5.2, allowing us to reconstruct non-perturbative physics out of perturbative data. However since these relations are very similar to the ones obtained in Section 5.2 we will not present them here.

The key message is that as soon as the number of chiral multiplets $r \in \mathbb{R}$ is kept generic, although the supersymmetry algebra is still formally respected, we have that all the perturbative series appearing in (95) are just asymptotic expansions. At this point we can make use of resurgent analysis to extract from the purely perturbative data non-perturbative information and only at the very end send the parameter $r \to N \in \mathbb{N}$ obtaining precisely the $\mathbb{CP}^{N-1}$ model result.

# 7 Conclusions

In this paper we consider the $\mathrm{S}^2$ partition function of the supersymmetric $\mathbb{CP}^{N-1}$ computed using localization and checked that we can reconstruct the expected chiral ring structure. The weak coupling expansion of this observable can be decomposed according to the resurgence triangle [44] and in each topological sector we find a perturbative series that truncates after finitely many orders making it seemingly impossible to exploit the resurgence machinery to reconstruct non-perturbative physics out of perturbative data. To this end we introduce, after having localized the path integral, a non-supersymmetric deformation that amounts to an unequal number of bosons and fermions. With this deformation in place we can reconstruct the full transseries representation of the deformed partition function and check that perturbation

theory does indeed become asymptotic. This is an example of Cheshire cat resurgence. We can use resurgent analysis to reconstruct from perturbative data the entire non-perturbative sectors previously completely hidden. Once we remove the deformation parameter we go back to the original undeformed case but we can still see the presence of resurgence at work.

Similarly we also consider a supersymmetry preserving deformation where we modify the number of chiral fields from $N \rightarrow r \in \mathbb{R}$ and study the $\mathbb{CP}^{r-1}$ model via analytic continuation. Although formally we still retain supersymmetry we immediately generate asymptotic transseries whenever $r$ is kept generic. We show that also in this case from the perturbative asymptotic series we can reconstruct the full transseries via resurgent analysis and only at the very end we send $r \rightarrow N \in \mathbb{N}$ to recover the $\mathbb{CP}^{N-1}$ result for which in each topological sector all the perturbative series truncate after finitely many orders.

This 2-dimensional example sheds some light on the role that resurgence plays in quantum field theories with convergent perturbative expansions. As in quantum mechanical examples [33, 34] also in here we can immediately see that a full transseries is hiding behind the "deceptive" convergent supersymmetric result as soon as an appropriate deformation is implemented. This $\Delta$ deformation we introduce is not fully satisfactory as it is not a genuine path integral deformation but rather corresponds to a mismatch between the number of bosons and fermions only *after* having localized the path integral. It would be interesting to see if a similar result can be obtained from a bona-fide deformation of the original path integral and perhaps understand how it relates to the thimble decomposition discussed in [60].

An interesting question would be to study the large-$N$ expansion of the $\mathbb{CP}^{N-1}$ partition function. It is not clear how the resurgence properties discussed in [43, 44] would arise from localization in the large-$N$ limit and what role the deformation has to play. Furthermore once the large-$N$ limit is computed we would like to understand, perhaps using similar methods to the one introduced in [61], how to interpolate this result with the finite $N$ case discussed in the present paper. It would also be interesting to understand how this large-$N$ limit is attained whether from taking $N$ over the natural numbers or over the reals since for finite $N$ the resurgence properties change dramatically as shown in this paper.

Although not fully satisfactory, the same type of $\Delta$ deformation can surely be implemented in basically all the supersymmetric localized theories. For example if we compute the $S^4$ partition function of $\mathcal{N} = 4 \, SU(N)$ SYM via localization [62, 63], since both the one-loop determinant and the instanton factor are trivial [63], the partition function is simply given by a Gaussian matrix model so it would seem that resurgence does not play any role. It would be very interesting to see if the deformation introduced in the present paper can be used to "deconstruct" this "1" in $\mathcal{N} = 4$ similarly to what the authors of [33] did to deconstruct the "0" of a vanishing ground state energy to uncover a Cheshire cat resurgence structure.

# Acknowledgement

We are thankful to Sungjay Lee for initial work on earlier stages of this project and to Stefano Cremonesi, Gerald Dunne, Vasilis Niarchos and Mithat Unsal for useful discussions and comments on the draft.

# A $\zeta$-function regularisation

Infinite products of the form

$$\prod_{k=0}^{\infty} (k+a)^{f(k)} \tag{100}$$

arise naturally when computing one-loop determinants, with $f(k)$ representing the degeneracy of the $k^{th}$ eigenvalue $(k+a)$. A standard way to regularise these type of products is to rewrite them in terms of the logarithm of the above expression using

$$\prod_{k=0}^{\infty}(k+a)^{f(k)} = \exp\left(\sum_{k=0}^{\infty} f(k)\log(k+a)\right). \tag{101}$$

Let us specialise now to the case

$$\prod_{k=0}^{\infty}(k+a) = \exp\left(\sum_{k=0}^{\infty} \log(k+a)\right), \tag{102}$$

which can be formally written as

$$\prod_{k=0}^{\infty}(k+a) = \exp\left(-\partial_s \zeta(s,a)|_{s=0}\right), \tag{103}$$

where $\zeta(s,a)$ denotes the Hurwitz-zeta function which is defined for complex arguments $s$ with $\text{Re}(s) > 1$ and $a$ with $\text{Re}(a) > 0$ via the series

$$\zeta(s,a) = \sum_{n=0}^{\infty} \frac{1}{(n+a)^s}, \tag{104}$$

and can be then extended to a meromorphic function defined for all $s \neq 1$. In particular one can show (for a short proof see [64])

$$\zeta'(0,a) - \zeta'(0) = \log\Gamma(a), \tag{105}$$

where $\zeta'(0) = d\zeta(s)/ds|_{s=0} = -\log\sqrt{2\pi}$ is the derivative of the Riemann-zeta at the origin. We can then rewrite a regularised version of the infinite product

$$\prod_{k=0}^{\infty}(k+a)\text{“} = \text{”}\frac{\sqrt{2\pi}}{\Gamma(a)}. \tag{106}$$

We need another regularised infinite product where the degeneracy $f(k)$ grows linearly with $k$, i.e. $f(k) = k+a$. We consider

$$\prod_{k=0}^{\infty}(k+a)^{k+a} = \exp\left(\sum_{k=0}^{\infty}(k+a)\log(k+a)\right) = \exp\left(-\partial_s \zeta(s,a)|_{s=-1}\right); \tag{107}$$

we need then a formula for $\partial_s \zeta(s,a)|_{s=-1}$, see [65,66].

We can proceed by first writing the asymptotic form (see http://dlmf.nist.gov/25.11.44 or [67])

$$\zeta'(-1,a) = \frac{1}{12} - \frac{a^2}{4} + \log a\left(\frac{1}{12} - \frac{a}{2} + \frac{a^2}{2}\right) - \sum_{k=1}^{\infty}\frac{B_{2k+2}}{(2k+2)(2k+1)2k}a^{-2k}, \tag{108}$$

with $B_n$ the Bernoulli numbers. By taking the derivative with respect to $a$ we obtain

$$\frac{\partial}{\partial a}\zeta'(-1,a) = a - \frac{1}{2} + \log\Gamma(a) + \zeta'(0), \tag{109}$$

which upon integration gives us the desired formula

$$\zeta'(-1,a) - \zeta'(-1) = \frac{1}{2}a(a-1) + a\zeta'(0) + \psi^{(-2)}(a), \tag{110}$$

where $\zeta'(-1) = d\zeta(s)/ds|_{s=-1} = 1/12 - \log G$ and $G$ denotes Glaisher constant $G = 1.282...$, while $\psi^{(-2)}(a) = \int da \log \Gamma(a)$. We can then rewrite a regularised version of the infinite product

$$\prod_{k=0}^{\infty} (k+a)^{k+a} \text{"}= \text{"} \exp\left(-\zeta'(-1) - \frac{1}{2}a(a-1) - a\zeta'(0) - \psi^{(-2)}(a)\right). \tag{111}$$

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
