# Peer review of "The grin of Cheshire cat resurgence from supersymmetric localization"

_SciPost Physics, doi:SciPost Phys. 4, 012 (2018)_

## Round 2 · Referee Report · Anonymous (Referee 1) · 2017-12-27

Strengths

  1. First example of Cheshire Cat Resurgence in quantum field theory
  2. Precise understanding of a relation between exact result and resurgence
  3. We can find relations among coefficients of pertubative expansions in different sectors
  4. Some interesting byproducts such as similar equation to topological-anti-topological equation for non-conformal case and truncation of perturbative expansion even when $\Delta$ is non-zero integer

Weaknesses

For the non-supersymmetric deformation, its interpretation from path integral is unclear.

Report

Dear Editors,

Resurgence approach is the poweful method to resum asymptotic perturbative series even if pertubative series around each saddle point has Borel ambiguities.
Typically cancellations of Borel ambiguities give relations among perturbative coefficients around different saddle points.

Sometimes we encounter a situation where perturbative series around trivial saddle point is truncated at some finite order and non-perturbative corrections are non-zero.
Probably most famous example for this is ground state energy of supersymmetric quantum mechanics whose supersymmetry is broken by non-perturbative effects.
For this case, it seems at first sight that there are no relations among coefficients around different saddle points from view point of resurgence since truncated series is completely unambiguous and naive application of resurgence to this case gives nothing.
However, last year, there appeared a notion of Cheshire Cat Resurgence in supersymmetric quantum mechanics with this type of situations.
In the papers last year, they considered supersymmetry breaking deformation of the quantum mechanics which makes perturbative series asymptotic.
For the deformed case, we can apply resurgence in a standard way and find relations between perturbative and non-perturbative data as usual.
A very interesting thing is that we still have the relations in undeformed limit.
Then it is natural to ask if similar mechanism holds also for quantum field theory.

This paper provides a first example of Cheshire Cat Resurgence in quantum field theory.
The authors focus on partition function of $\mathcal{N}=(2,2)$ supersymmetric $\mathbf{CP}^{N-1}$ model on (B-type) $S^2$
whose exact formula is available thanks to (Coulomb branch) localization and given by a one-dimensional integral.
Then they study large FI parameter expansion of the 1d integral.
It turns out that the partition function receives non-pertuabative corrections and perturbative series in every sector is truncated at finite orders.
Therefore naively applying resurgence to this problem does not give nothing.

In this situation, the authors consider two types of deformation of the 1d integral for the partition function which make the perturbative series asymptotic.
First, they introduce an unbalance of powers of one-loop determinants in the localization coming from bosonic and fermionic degrees of freedom.
Then they demonstrate Cheshire cat resurgence scenario for the deformed 1d integral and it quite works though its path integral interpretation is unclear.
Especially they find relations between perturbative and non-perturbative data even after turning off the deformation.
Second, they consider analytic continuation of $N$ to a complex number in the 1d integral.
Then they found that perturbative series becomes asymptotic for non-integer $N$ and Cheshire cat resurgence structure also for this deformation.

Besides these, there are some interesting byproducts in this paper such as similar equation to topological-anti-topological equation for non-conformal case
and truncation of perturbative expansion even when $\Delta$ is non-zero integer.

I think that this paper is well-written and gives very important insights to resurgence structure of quantum field theory.
Thus I strongly recommend this paper for publication in SciPost Physics after modifying some minor typos.

Best

Requested changes

Typos: 1. In second term of (2.16), I think that $e^{-4\pi\xi}$ is missing. 2. In second line of page 26, "." is missing. 3. In the second paragraph of page 26, they refer eq.(6.8) but I guess that they would like to refer another equation.

  • validity: top
  • significance: top
  • originality: high
  • clarity: high
  • formatting: excellent
  • grammar: perfect

Author:  Daniele Dorigoni  on 2018-01-10  [id 198]

(in reply to Report 1 on 2017-12-27)
Category:
correction

We thank the referee for the time spent reading our paper and for the nice comments.
We completely agree with the typos pointed out.

---

## Round 3 · Author Response

We thank the editor and the referee for their time and work.

---

## Round 3 · List of Changes

We have addressed all the minor typos pointed out by the referee.

On page 10, equation (2.16) we added the factor $e^{-4 \pi \xi} $ multiplying the second term

Second line on page 26 we added the missing "."

In the second paragraph of page 26, we corrected the reference from eq. (6.8) to eq. (5.1)

You are currently on this page

Resubmission 1711.04802v3 on 11 January 2018

---

## Editorial Decision

published